# MKK6 deficiency promotes cardiac dysfunction through MKK3-p38γ/δ-mTOR hyperactivation

**Rafael Romero-Becerra[1], Alfonso Mora[1], Elisa Manieri[1], Ivana Nikolic[1], Ayelén Melina Santamans[1], Valle Montalvo-Romeral[1], Francisco Miguel Cruz[1], Elena Rodríguez[1], Marta León[1], Luis Leiva-Vega[1], Laura Sanz[1], Víctor Bondía[1], David Filgueiras-Rama[1,2,3], Luis Jesús Jiménez-Borreguero[1], José Jalife[1,2,4], Barbara Gonzalez-Teran[1,5]\*, Guadalupe Sabio[1]\***

[1]Centro Nacional de Investigaciones Cardiovasculares, Madrid, Spain; [2]CIBER de Enfermedades Cardiovasculares, Madrid, Spain; [3]Hospital Clínico Universitario San Carlos, Madrid, Spain; [4]Center for Arrhythmia Research, Department of Internal Medicine, University of Michigan, Ann Arbor, Ann Arbor, United States; [5]Gladstone Institutes, San Francisco, United States

**Abstract** Stress-activated p38 kinases control a plethora of functions, and their dysregulation has been linked to the development of steatosis, obesity, immune disorders, and cancer. Therefore, they have been identified as potential targets for novel therapeutic strategies. There are four p38 family members (p38α, p38β, p38γ, and p38δ) that are activated by MKK3 and MKK6. Here, we demonstrate that lack of MKK6 reduces the lifespan in mice. Longitudinal study of cardiac function in MKK6 KO mice showed that young mice develop cardiac hypertrophy which progresses to cardiac dilatation and fibrosis with age. Mechanistically, lack of MKK6 blunts p38α activation while causing MKK3-p38γ/δ hyperphosphorylation and increased mammalian target of rapamycin (mTOR) signaling, resulting in cardiac hypertrophy. Cardiac hypertrophy in MKK6 KO mice is reverted by knocking out either p38γ or p38δ or by inhibiting the mTOR pathway with rapamycin. In conclusion, we have identified a key role for the MKK3/6-p38γ/δ pathway in the development of cardiac hypertrophy, which has important implications for the clinical use of p38α inhibitors in the long-term treatment since they might result in cardiotoxicity.

*For correspondence: barbara.gonzalezteran@ gladstone.ucsf.edu (BG-T); gsabio@cnic.es (GS)

**Competing interest:** The authors declare that no competing interests exist.

## Editor's evaluation

The paper for the first time shows that MKK6 reduces life span in mice, and that young mice with MKK deficiency display cardiac hypertrophy and that it progresses to cardiac dilatation and fibrosis as they age. The paper also provides a mechanism for this phenomenon and demonstrate that MKK6 deletion leads to reduced p38a activation but at the same time causes MKK3-p38g/d hyperphosphorylation and increased mTOR signaling. These studies are novel and will advance our understanding of the role of this pathway in aging.

## Introduction

Cardiac hypertrophy is an adaptive response of the heart to hemodynamic stress that can be physiological (e.g. pregnancy or exercise) or pathological (e.g. hypertension or valvular disease). Physiological cardiac hypertrophy is accompanied by a normal or even enhanced cardiac function, while pathological forms of hypertrophy are accompanied by myocardial dysfunction and fibrosis and

**eLife digest** The human heart can increase its size to supply more blood to the body's organs. This process, called hypertrophy, can happen during exercise or be caused by medical conditions, such as high blood pressure or inherited genetic diseases. If hypertrophy is continually driven by illness, this can cause the heart to fail and no longer be able to properly pump blood around the body.

For hypertrophy to happen, several molecular changes occur in the cells responsible for contracting the heart, including activation of the p38 pathway. Within this pathway is a p38 enzyme as well as a series of other proteins which are sequentially turned on in response to stress, such as inflammatory molecules or mechanical forces that alter the cell's shape.

There are different types of p38 enzyme which have been linked to other diseases, making them a promising target for drug development. However, clinical trials blocking individual members of the p38 family have had disappointing results. An alternative approach is to target other proteins involved in the p38 pathway, such as MKK6, but it is not known what effect this might have.

To investigate, Romero-Becerra et al. genetically modified mice to not have any MKK6 protein. As a result, these mice had a shorter lifespan, with hypertrophy developing at a young age that led to heart problems. Romero-Becerra et al. used different mice models to understand why this happened, showing that a lack of MKK6 reduces the activity of a specific member of the p38 family called p38α. However, this blockage boosted a different branch of the pathway which involved two other p38 proteins, p38γ and p38δ. This, in turn, triggered another key pathway called mTOR which also promotes hypertrophy of the heart.

These results suggest that drugs blocking MKK6 and p38α could lead to side effects that cause further harm to the heart. A more promising approach for treating hypertrophic heart conditions could be to inhibit p38γ and/or p38δ. However, before this can be fully explored, further work is needed to generate compounds that specifically target these proteins.

represent a risk factor for ventricular arrhythmias and sudden cardiac death (*Maillet et al., 2013*; *Nakamura and Sadoshima, 2018*; *Oldfield et al., 2020*).

Initially, cardiac hypertrophy is induced as a compensatory response to preserve cardiac function under stressful conditions, a process known as adaptive cardiac hypertrophy. However, if the pathological stimulus is maintained, this adaptive cardiac hypertrophy will eventually lead to the development of pathological cardiac hypertrophy and heart failure (*Nakamura and Sadoshima, 2018*; *Oldfield et al., 2020*). The form of cardiac hypertrophy developed will depend on the type of the hypertrophic stimuli, the duration of the stimuli, and the downstream signaling involved (*Nakamura and Sadoshima, 2018*; *Oldfield et al., 2020*; *Shimizu and Minamino, 2016*). Several signaling pathways known to promote physiological cardiac hypertrophic growth have been found to drive pathological hypertrophy and cardiac dysfunction when persistently activated (*Heineke and Molkentin, 2006*; *Maillet et al., 2013*; *Nakamura and Sadoshima, 2018*; *Porrello et al., 2008*). For instance, IGF1 or Akt transgenic mice develop proportionately enlarged hearts with initially normal cardiac function, which over time progress to pathological hypertrophy with impaired cardiac function (*Delaughter et al., 1999*; *Shiojima et al., 2005*). An essential feature of both physiological and pathological hypertrophy is increased protein synthesis, critically regulated by the mammalian target of rapamycin (mTOR) pathway mainly through the phosphorylation of its downstream substrates. Activation of mTOR signaling is increased during postnatal cardiac development (*González-Terán et al., 2016*) as well as in the hearts of transgenic mouse models suffering from physiological cardiac hypertrophy (*McMullen et al., 2004a*; *McMullen et al., 2004b*; *Shioi et al., 2000*; *Shioi et al., 2003*). Moreover, the specific mTOR inhibitor rapamycin attenuates and reverses cardiac-overload-induced pathological hypertrophy (*Shioi et al., 2003*).

Stress-inducing stimuli in the heart activate several mitogen-activated protein kinases (MAPKs) including the p38 family. p38 kinases control a wide range of processes, and their dysregulation has been linked to numerous diseases, making them a promising pharmacological target for therapeutic use (*Canovas and Nebreda, 2021*). The main p38 upstream activators are the MAPK MKK3 and MKK6, which are highly selective for p38 MAPKs and do not activate c-Jun N-terminal kinases (JNKs) or ERK1/2 (*Cuenda et al., 1996*; *Cuenda et al., 1997*; *Dérijard et al., 1995*). Mice lacking both MKK3

and MKK6 are non-viable and die in mid-gestation, associated with defects in the placenta and the embryonic vasculature (*Brancho et al., 2003*). In contrast, mice individually lacking MKK3 or MKK6 are healthy, indicating partial functional overlap (*Lu et al., 1999*; *Tanaka et al., 2002*; *Wysk et al., 1999*). Both MKKs are expressed in cardiac tissue and have been implicated in cardiac hypertrophy. But, this evidence is based on overexpressing dominant negative or constitutively active MKK3 and MKK6 mutants, and these non-physiological models could produce artifactual phenotypes (*Braz et al., 2003*; *Muslin, 2008*). Moreover, it is not known which MKK is the main isoform required for activation of the different p38s family members (p38α, p38β, p38γ, and p38δ) during postnatal growth and in response to other hypertrophic stimuli. The heart expresses all four p38s, but protein levels are higher for p38α and p38γ (*Dumont et al., 2019*). During embryonic development, p38α and p38β play redundant roles in the regulation of cardiac proliferation; mice lacking both kinases develop septal defects. However, mice lacking cardiac p38α or p38β alone have a normal cardiac structure and function under baseline conditions (*del Barco Barrantes et al., 2011*). In response to cardiac overload, p38α contributes to myocyte apoptosis (*Nishida et al., 2004*), whereas p38β might affect hypertrophic response (*Baines and Molkentin, 2005*; *Braz et al., 2003*; *Liao et al., 2001*; *Petrich and Wang, 2004*; *Wang, 2007*). We have previously shown that p38γ and p38δ have a predominant role in the regulation of the cardiac hypertrophy mediating physiological early postnatal and pathological angiotensin II-induced cardiac hypertrophy through mTOR activation (*González-Terán et al., 2016*). Additionally, p38γ and p38δ control the postnatal metabolic switch in the heart blocking glycogen storage by the inhibition of GYS1 (*Santamans et al., 2021*).

Here, we demonstrate a not previously described specificity of MKK3 for the activation of p38γ/δ and MKK6 for the activation of p38α in the heart. Furthermore, we find that MKK3/p38γ/δ hyperactivation in MKK6 KO mice promotes cardiac hypertrophy through mTOR activation, which progresses to a pathological cardiac hypertrophy phenotype with age and development of cardiac dysfunction. In addition, using mice lacking muscle p38α, we found that this kinase inhibits MKK3 expression and activation in the heart. Our results have important implications for the clinical use of p38α inhibitors in long-term treatment since they might result in hyperactivation of MKK3/p38γ/δ axis triggering cardiotoxicity.

## Results

### MKK6-deficient mice die prematurely

Several studies have addressed the role of p38 signaling in homeostasis and disease (*Nikolic et al., 2020*; *Romero-Becerra et al., 2020*). While mice lacking both MKK3 and MKK6 die in mid-gestation with mutant embryos demonstrating abnormalities of the placenta and embryonic vasculature (*Brancho et al., 2003*), mice individually lacking MKK3 or MKK6 are viable and fertile with no reported developmental abnormalities, suggesting partial functional redundancy of these kinases (*Lu et al., 1999*; *Tanaka et al., 2002*; *Wysk et al., 1999*). However, the role of the p38 pathway in aging remains incompletely understood. Therefore, we examined mice harboring germline deletion of MKK6 (MKK6 KO) (*Tanaka et al., 2002*) at an advanced age. Mice lacking MKK6 have reduced body weight compared to age-matched wild type (WT) animals (*Figure 1A and B*), which can be partially explained by a dramatic reduction in white adipose tissue (*Figure 1C*). This agrees with previous studies demonstrating that MKK6 KO mice are protected against diet-induced obesity with increased browning of the epididymal white adipose tissue (*Matesanz et al., 2017*).

Additionally, these mice exhibit an abnormal posture characterized by a hunched position and the development of thoracic kyphosis and severe ataxia (*Figure 1D*, *Figure 1—video 1*). As a consequence of all these phenotypic alterations, MKK6 KO mice suffer premature death, the first mice dying at 51 weeks of age with a median lifespan of 76 weeks (*Figure 1E*).

### MKK6-deficient mice develop increased age-related cardiac dysfunction

The downstream kinases of MKK6 have been implicated in major cardiovascular abnormalities during development. Combined deletion of p38α and p38β results in cardiac defects during embryonic development (*del Barco Barrantes et al., 2011*), whereas p38γ/δ deficient mice exhibit reduced cardiomyocyte hypertrophic growth and smaller hearts (*González-Terán et al., 2016*), and cardiovascular disease is a common characteristic in premature aging syndromes (*Carrero et al., 2016*).

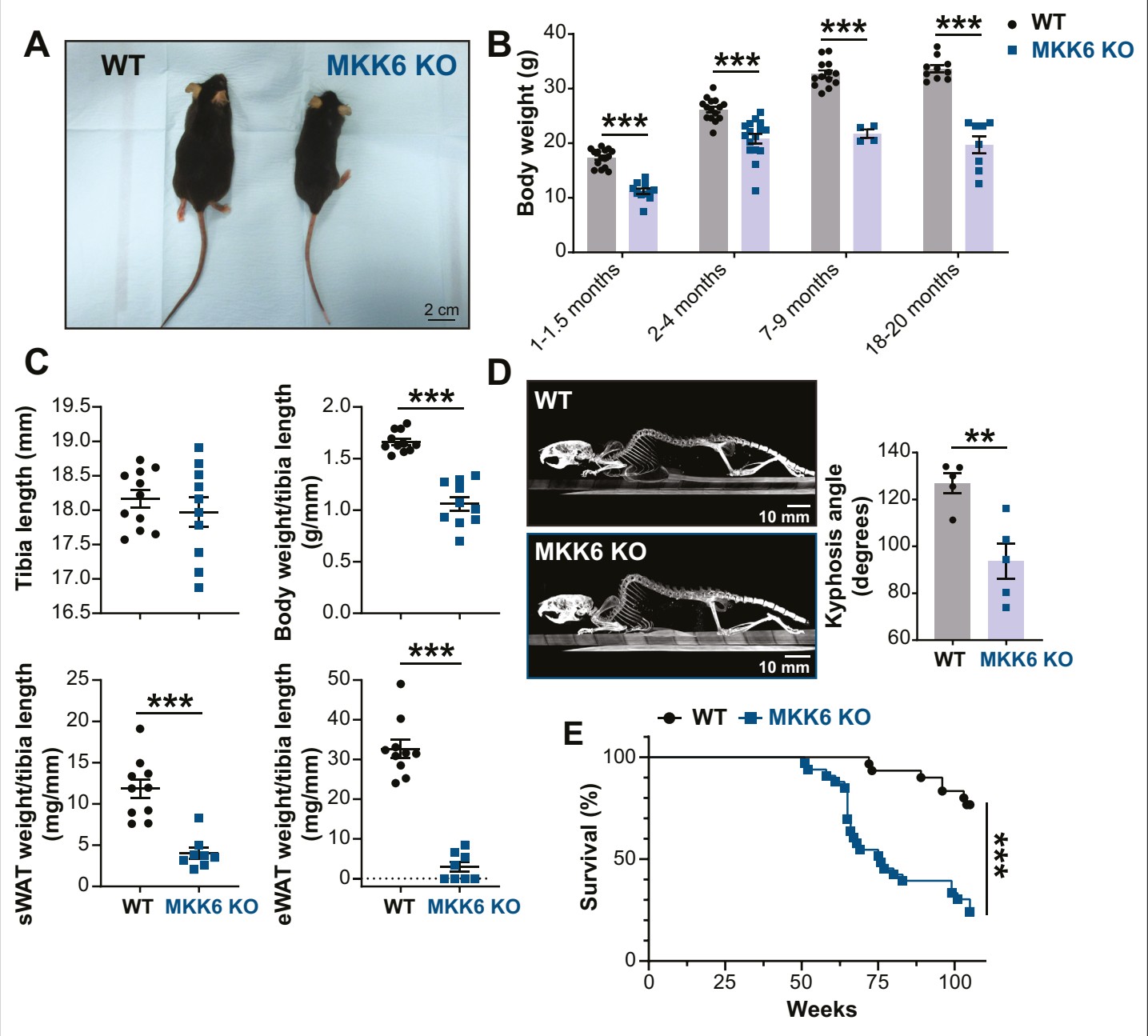

**Figure 1.** MKK6 KO mice show a reduced survival age. (**A**) Representative picture of 19-month-old wild type (WT) and MKK6 KO male mice. Scale bar: 2 cm. (**B**) Body weight of WT (n=10–15) and MKK6 KO (n=4–16) mice over the indicated age period. Two-way ANOVA followed by Sidak's post-test. (**C**) Tibia length and body weight, subcutaneous white adipose tissue (sWAT) and epidydimal white adipose tissue (eWAT) to tibia length ratio from 20-month-old WT (n=10–11) and MKK6 KO (n=8–10) mice. Unpaired *t*-test or Mann-Whitney U test. (**D**) Representative CT scan images and quantification of the column kyphosis angle of 19-month-old WT (n=5) and MKK6 KO (n=5) mice. Unpaired *t*-test. Scale bar: 10 mm. (**E**) Kaplan-Meier survival plot of age-related mortality in WT (n=30) and MKK6 KO (n=33) mice. An endpoint of 105 weeks was chosen to avoid a severe worsening of the mice's health. Gehan-Breslow-Wilcoxon test. Data in B–D are mean ± SEM. **p<0.01; ***p<0.001.

The online version of this article includes the following video and source data for figure 1:

**Source data 1.** Raw data, statistical tests and significance, and *n* for *Figure 1B, C and D & E*.

**Figure 1—video 1.** Ataxia and hunched posture in MKK6 KO mice.

https://elifesciences.org/articles/75250/figures#fig1video1

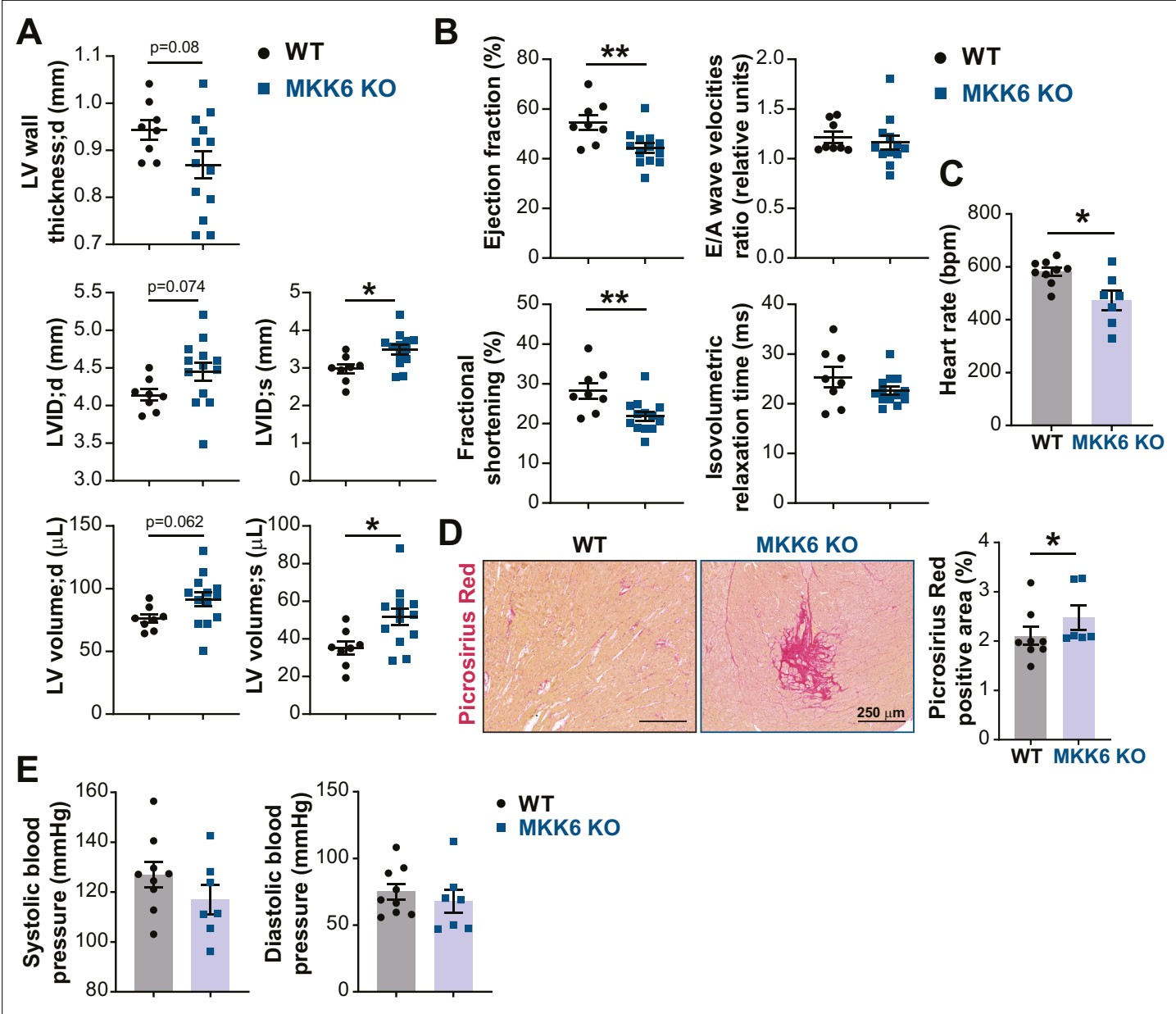

**Figure 2.** MKK6 deficiency promotes cardiac dysfunction at advanced ages. (**A** and **B**) Echocardiography parameters related to left ventricle (LV) dimensions (**A**) and contractility (**B**) in 12–14-month-old wild type (WT) (n=8) and MKK6 KO (n=13). Each dot corresponds to an individual animal. LV wall thickness;d (LV wall thickness in diastole), LVID;d (left ventricular internal diameter in diastole), LVID;s (left ventricular internal diameter in systole), LV volume;d (left ventricular volume in diastole), and LV volume;s (left ventricular volume in systole). Unpaired *t*-test. (**C**) Heart rate in conscious 18-month-old WT (n=9) and MKK6 KO (n=7) mice. bpm (beats per minute). Unpaired *t*-test. (**D**) Picrosirius red staining and quantification of cardiac fibrosis in 23–24-month-old WT (n=8) and MKK6 KO (n=6) mice. Mann-Whitney U test. Scale bars: 250 µm. (**E**) Systolic and diastolic blood pressure measured in conscious 18-month-old WT (n=9) and MKK6 KO (n=7) mice. Unpaired *t*-test. Data in A–E are mean ± SEM. *p<0.05; **p<0.01.

The online version of this article includes the following source data for figure 2:

**Source data 1.** Raw data, statistical tests and significance, and *n* for *Figure 2*.

This prompted us to speculate that cardiac abnormalities could be one of the underlying causes of premature death of MKK6 KO mice. Echocardiographic analyses of 12–14-month-old mice demonstrated eccentric hypertrophy in MKK6 KO mice compared to control mice, as detected by thinning of the left ventricle (LV) wall, as well as increased left ventricular internal diameter and left ventricular volume, especially during the systole (*Figure 2A*). Cardiac enlargement compromised systolic function, evidenced by a decrease in the ejection fraction and fractional shortening. However, the diastolic

function appeared to be maintained, with a normal E/A wave velocities ratio and isovolumetric relaxation time (*Figure 2B*). Moreover, MKK6 KO mice exhibit bradycardia (*Figure 2C*). We performed picrosirius red staining for collagen and quantified positive areas in serial histologic sections from MKK6 KO and WT hearts and found cardiac fibrotic lesions in MKK6 KO old mice (*Figure 2D*). To discard hypertension as a possible contributor to the cardiac dysfunction, we evaluated blood pressure in these animals. MKK6 KO mice did not present differences in blood pressure compared to age-matched controls (*Figure 2E*).

## Young MKK6-deficient mice present cardiac hypertrophy

Cardiac dysfunction may result from an initial compensated cardiac hypertrophy that with time becomes pathological (*Nakamura and Sadoshima, 2018*). Echocardiographic analysis at 9 weeks of age demonstrated cardiac hypertrophy in MKK6-deficient animals when compared with controls, as detected by measures of left ventricular mass, interventricular septal thickness, left ventricular posterior wall thickness, and left ventricular internal diameter (*Figure 3A*). However, cardiac enlargement did not compromise systolic or diastolic function (given by the ejection fraction and the E/A wave velocity ratio, respectively) but was accompanied by an increase in stroke volume and cardiac output (*Figure 3B*). Gross anatomic and histologic analyses confirmed that these non-invasive findings as MKK6-deficient hearts were larger than WT controls when normalized to tibia length (TL) (*Figure 3C&D*), a difference that was not apparent at 4 weeks of age. Serial analysis of heart weight (HW)/TL over 15 weeks demonstrated enhanced cardiac growth in MKK6 KO mice (*Figure 3D*). The progressive increase in the size of MKK6 KO hearts correlated with increased cardiomyocyte cross-sectional area, consistent with enhanced hypertrophic growth (*Figure 3E*). Hypertension was excluded as a possible contributor to this increased growth as MKK6 KO mice demonstrate reduced systolic blood pressures when compared to age-matched controls (*Figure 3—figure supplement 1A*). Fibrosis and reactivation of a 'fetal gene program' (e.g. *Nppa*, *Nppb*, *Acta2*, and *Myh7*) are hallmark features of pathological hypertrophy (*Bernardo et al., 2010*). Histological cardiac examination revealed no evidence of fibrosis in MKK6 KO heart sections (*Figure 3—figure supplement 1B*). We also found no difference in the expression of fibrotic genes including *Col1a1*, *Col3a1, and Fn* or markers of the fetal gene program (*Figure 3—figure supplement 1C,D*). No meaningful changes were visible for *Nppa* or *Nppb* (*Figure 3—figure supplement 1D*). These observations suggest that at 9 weeks of age, MKK6 KO hearts show a non-pathological cardiac hypertrophy.

To confirm the MKK6 autonomous effect in cardiomyocytes, we employed a murine conditional MKK6 allele (*Map2k6^LoxP^)^23^* and two cardiomyocyte Cre-expressing lines *Mck-Cre* (*Brüning et al., 1998*) and *Myh6-Cre* (αMHC-Cre) (*McFadden et al., 2005*) to assess the consequences of MKK6 genetic ablation in postnatal cardiomyocytes. MCK-Cre is active in striated muscle, and the *Mck-Cre; Map2k6^LoxP/LoxP^* (MKK6^MCK−KO^) mice demonstrated specific and efficient deletion of MKK6 in the heart and skeletal muscle tissues, but not in spleen or liver (*Figure 4—figure supplement 1A,B*). Importantly, the cardiac phenotype of 9-weeks-old MKK6^MCK−KO^ mice resembled that of MKK6 KO animals (*Figure 4A–C*). Similar results were obtained with *Myh6-Cre; Map2k6^LoxP/LoxP^* (MKK6^αMHC−KO^) mice (*Figure 4D–F*). These data collectively confirm that cardiomyocyte MKK6 controls heart growth. Finally, to corroborate that cardiac hypertrophy caused by MKK6 deficiency in cardiomyocytes induce premature death and associated cardiac dysfunction, we evaluated the lifespan and cardiac function at advanced ages in MKK6^αMHC−KO^. Similar to the whole-body MKK6 KO animals, MKK6^αMHC−KO^ mice showed a reduced survival, with a median lifespan of 51 weeks (*Figure 4—figure supplement 2A*). This increased mortality was accompanied by cardiac dilatation and cardiac dysfunction, as revealed by echocardiographic analysis (*Figure 4—figure supplement 2B and C*).

## MKK6-deficient hearts have increased MKK3-p38γ/δ activation

MKK6 is a critical upstream activator of p38 MAPKs, but its specificity for individual p38 family members is not well established. We assessed the relative levels of phosphorylated p38 isoforms by immunoprecipitation in MKK6 KO hearts, which demonstrated hyperphosphorylation of p38γ and p38δ (*Figure 5A*) with a simultaneous reduction in phosphorylation of p38α (*Figure 5B*). Immunoblot analysis also revealed increased levels of phosphorylated MKK3, the other main p38 upstream activator, in MKK6 KO hearts (*Figure 5C*). This observation suggested that increased phosphorylation of p38γ and p38δ in the MKK6 KO hearts resulted from MKK3 activation. Accordingly, phosphorylation of

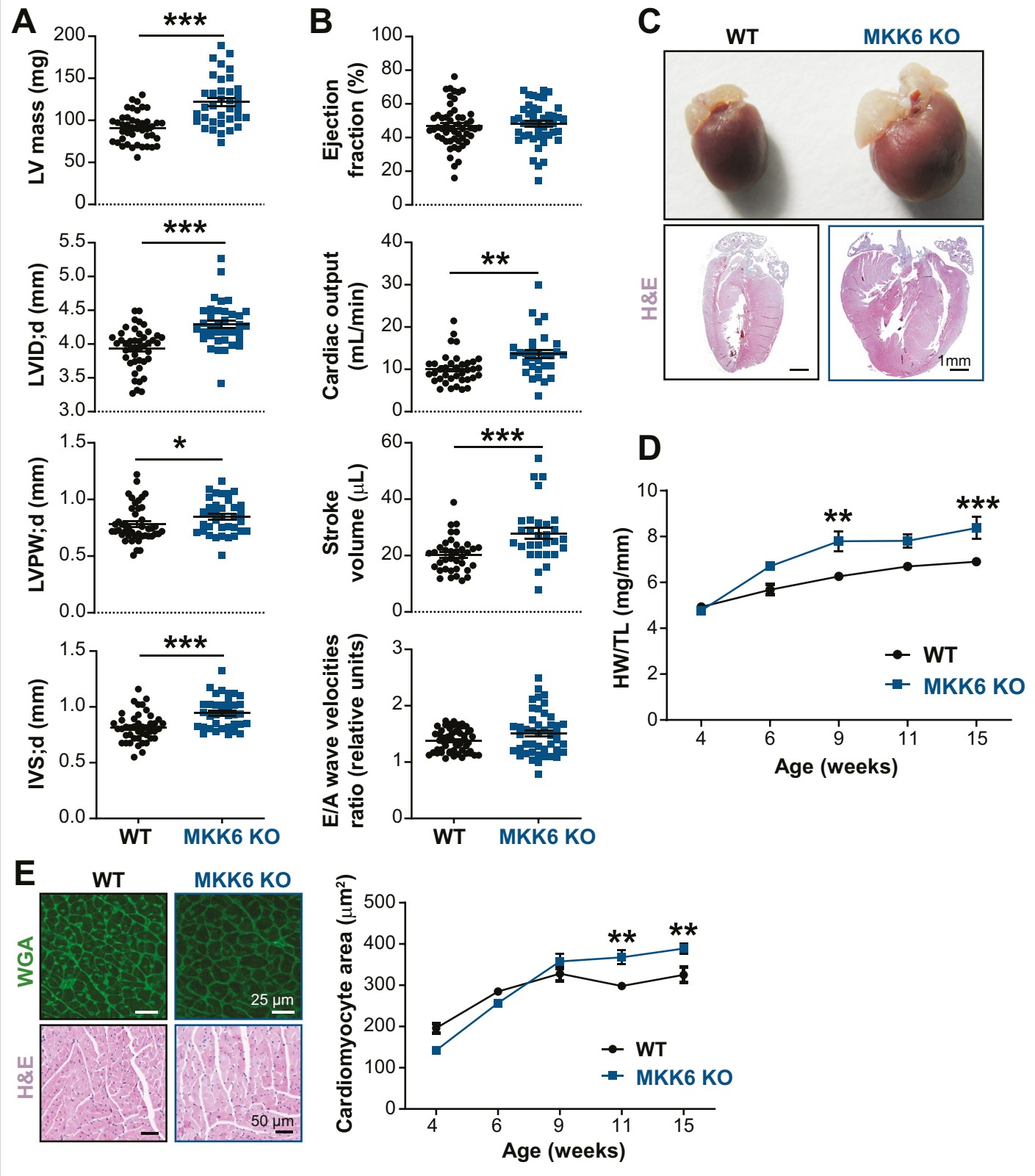

**Figure 3.** Young MKK6-deficient hearts are hypertrophic with preserved cardiac function. (**A** and **B**) Echocardiography parameters related to left ventricle (LV) dimensions (**A**) and contractility (**B**) in 9-week-old wild type (WT) (n=37–54) and MKK6 KO (n=29–46) mice. Each dot corresponds to an individual mouse. Mean ± SEM are shown as well. LV mass (left ventricular mass), LVID;d (left ventricular internal diameter in diastole), LVPW;d (left ventricular posterior wall in diastole), and IVS;d (inter-ventricular septum in diastole). Unpaired *t*-test or Mann-Whitney U test. (**C**) Representative whole

*Figure 3 continued on next page*

*Figure 3 continued*

hearts and cardiac longitudinal sections stained with hematoxylin and eosin (H&E) from 9-week-old WT and MKK6 KO mice. Scale bars: 1 mm. (**D**) Heart weight to tibia length ratio (HW/TL) of WT (n=4–15) and MKK6 KO (n=5–14) over the indicated age period. Two-way ANOVA followed by Sidak's post-test. (**E**) Top: representative Fluorescein isothiocyanate (FITC) wheat germ agglutinin (FITC-WGA)-stained heart sections from 9-week-old WT and MKK6 KO mice and quantification of cardiomyocyte cross-sectional area over time (right graph, WT n=4–5; MKK6 KO n=4–6, and two-way ANOVA followed by Sidak's post-test). Scale bars: 25 µm. Bottom: representative H&E-stained heart sections. Scale bars: 50 µm. Data in A, B, D, and E are mean ± SEM. *p<0.05; **p<0.01; ***p<0.001.

The online version of this article includes the following source data and figure supplement(s) for figure 3:

**Source data 1.** Raw data, statistical tests and significance, and *n* for *Figure 3A, B and D&E* and *Figure 3—figure supplement 1*.

**Figure supplement 1.** Evaluation of hallmarks of pathological cardiac hypertrophy in young MKK6 KO mice.

p38γ and p38δ was strongly reduced in MKK3-deficient mice, whereas p38α phosphorylation was not changed appreciably (*Figure 5D* and *Figure 5—figure supplement 1A*). In addition, hearts of MKK3 KO mice were smaller at 9 weeks of age when compared with age-matched WT controls (*Figure 5E*), a finding consistent with the previously described roles of p38γ and p38δ in promoting postnatal cardiac hypertrophic growth (*González-Terán et al., 2016*). To evaluate if p38α decreased activity was involved in the MKK3 increased phosphorylation observed in MKK6 KO mice, we generated mice lacking p38α specifically in muscle (*Mck-Cre; Mapk14^{LoxP/LoxP}* [p38α^{MCK-KO}]). p38α^{MCK-KO} mice presented an increased expression and phosphorylation of MKK3 and MKK6 (*Figure 5—figure supplement 2A*). This agreed with previous evidence indicating that p38α regulates p38 upstream kinases at the transcriptional level (*Ambrosino et al., 2003*). We then evaluated MKK3 expression in hearts from MKK6 KO mice and found that lack of p38α activation also promoted an increase in MKK3 mRNA and protein levels in these animals (*Figure 5—figure supplement 2B and C*), indicating a negative feedback regulation of MKK3 expression and activity by p38α activation. Taken together, these observations suggest that in the heart, MKK6 primarily targets p38α and MKK3 targets p38γ and p38δ. In addition, they suggest that MKK6 deficiency leads to cardiac hypertrophy via activation of MKK3 and p38γ and p38δ.

## Hypertrophy in MKK6-deficient hearts is mediated by p38γ/δ

To confirm that enhanced hypertrophic growth in MKK6-deficient mice is mediated by modulation of p38γ/δ activation, we introduced a deletion of *Mapk12* (p38γ) in the context of MKK6 deficiency. The double mutant combination (MKK6/p38γ DKO) rescued normal cardiac growth, as the heart sizes and cardiomyocyte cross-sectional areas of the double mutants were equivalent to those of WT controls (*Figure 6A–C*). We further demonstrated the requirement for p38δ to be cell-autonomous in striated muscle using a p38δ conditional allele (*Mapk13^{LoxP}*). MKK6-deficient hearts lacking p38δ in their myocytes (MKK6 KO; *Mck-Cre; Mapk13^{LoxP/LoxP}* [MKK6 KO/p38δ^{MCK–KO}]) were similar in size to those of mice lacking p38δ in their myocytes (*Mck-Cre; Mapk13^{LoxP/LoxP}* [p38δ^{MCK–KO}]), with no appreciable increase in cardiomyocyte cross-sectional area (*Figure 6D–F*). Collectively, these data demonstrate that enhanced cardiac hypertrophic growth in MKK6-deficient mice is mediated by hyperactivation of p38γ/δ signaling in striated muscle.

## mTOR pathway hyperactivation mediated hypertrophy in MKK6-deficient hearts

The p38γ and p38δ isoforms have previously been demonstrated to promote cardiac hypertrophic growth through activation of mTOR signaling by targeting the mTOR-inhibitory protein DEPTOR for degradation (*González-Terán et al., 2016*). As expected from that finding, immunoblot analysis of MKK6 KO mouse hearts showed an increase in mTOR pathway activation (*Figure 7A*), together with reduced levels of DEPTOR protein (*Figure 7—figure supplement 1A*). As protein synthesis represents a key target of the mTOR signaling pathway and is critical for cardiomyocyte hypertrophic growth, we assessed eukaryotic initiation/elongation factors by immunoblot analyses. We found an overall increase in translational activation in MKK6 KO hearts relative to WT (*Figure 7B*). We corroborated these findings by analyzing puromycin incorporation into newly synthesized peptides in WT and MKK6 KO hearts, which demonstrated greater puromycin labeling (*Figure 7C*) and overall protein content (*Figure 7D*) in mutant hearts.

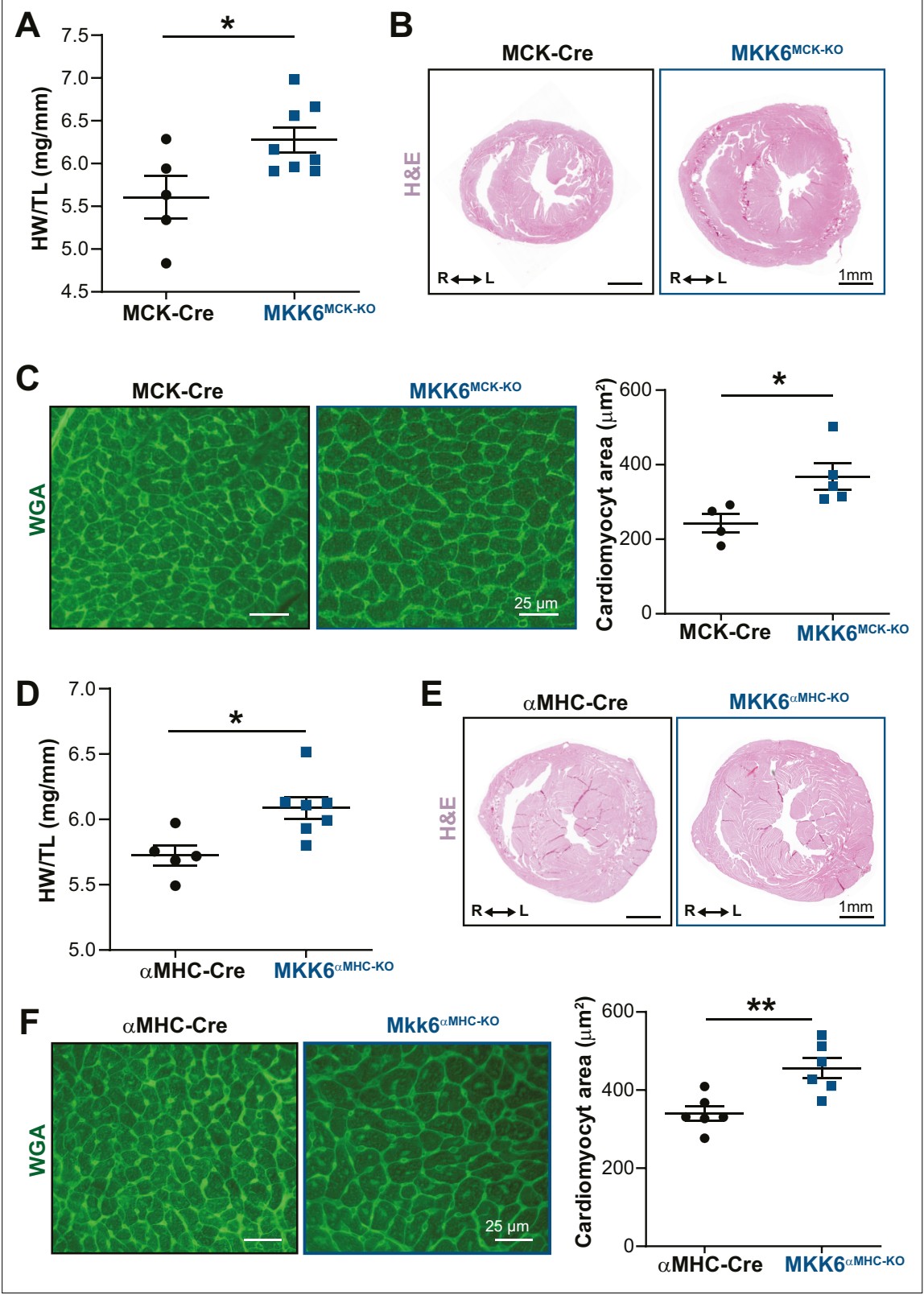

**Figure 4.** Cardiac MKK6 controls postnatal heart growth. (**A**) Heart weight to tibia length ratio (HW/TL) in 9-week-old MCK-Cre (n=5) and MKK6<sup>MCK-KO</sup> (n=8) mice. Unpaired *t*-test. (**B**) Hematoxylin and eosin (H&E)-stained transverse cardiac sections from control (MCK-Cre) and MKK6<sup>MCK-KO</sup> mice. Scale bars: 1 mm. (**C**) Representative FITC wheat germ agglutinin (FITC-WGA) staining and corresponding quantification of cardiomyocyte cross-sectional area in MCK-Cre (n=4) and MKK6<sup>MCK-KO</sup> (n=5) mice. Unpaired *t*-test. Scale bars: 25 μm. (**D**) HW/TL in 9-week-old αMHC-Cre (n=5) and MKK6<sup>αMHC-KO</sup> (n=7)

*Figure 4 continued on next page*

*Figure 4 continued*

mice. Unpaired *t*-test. (**E**) H&E-stained transverse heart sections from αMHC-Cre and MKK6^αMHC-KO mice. Scale bars: 1 mm. (**F**) Representative FITC-WGA staining and corresponding quantification of cardiomyocyte cross-sectional area in αMHC-Cre (n=6) and MKK6^αMHC-KO (n=6) mice. Unpaired *t*-test. Scale bars: 25 μm. Data in A, C, D, and F are mean ± SEM. *p<0.05; **p<0.01.

The online version of this article includes the following source data and figure supplement(s) for figure 4:

**Source data 1.** Raw data, statistical test and significance, and *n* for *Figure 4A, C, D and F* and *Figure 4—figure supplement 2*.

**Figure supplement 1.** Tissue-specific MKK6 deletion in MKK6^MCK-KO and MKK6^αMHC-KO mice.

**Figure supplement 1—source data 1.** Raw blots for *Figure 4—figure supplement 1*.

**Figure supplement 2.** MKK6^αMHC-KO mice show a reduced lifespan and associated cardiac dysfunction.

Examination of MKK6-deficient mouse embryonic fibroblasts (MEFs) yielded similar results to those observed in cardiac lysates. Namely, MKK6-deficient MEFs showed mTOR pathway activation and increased cell size (*Figure 7—figure supplement 2A and B*). Furthermore, lentiviral infection of MEFs with an active form of p38α (D176A/F327S) was able to reduce mTOR pathway activity as well as cell size (*Figure 7—figure supplement 2A and B*).

Given these results, we next examined the extent to which cardiac hypertrophy in MKK6 KO mice is mediated by increased mTOR signaling. We blocked mTOR activation by daily intraperitoneal injection of rapamycin, a potent and specific small molecule mTOR inhibitor, from 3 to 9 weeks of age. This treatment was sufficient to bring the heart size (HW/TL) of MKK6 KO animals close to that of WT age-matched controls, which corresponded with a robust reduction in cardiomyocyte cross-sectional area in MKK6 KO mice (*Figure 7E-G*), altogether suggesting that the cardiac hypertrophy in MKK6 KO mice results from hyperactivation of mTOR signaling.

To test whether MKK3 inhibition was able to attenuate mTOR activation and cardiac growth in MKK6 KO mice, we generated mice lacking both, MKK3 and MKK6, in striated muscle (*Mck-Cre; Map2k3^LoxP/LoxP/Map2k6^LoxP/LoxP*; MKK3/6^MCK-KO, *Figure 7—figure supplement 2*), since whole-body MKK3 and MKK6 double KO mice are non-viable (*Brancho et al., 2003*). We found that MKK3 deletion in the context of MKK6 deficiency promoted a reduction in the phosphorylation of p38γ as well as in mTOR pathway activity (*Figure 7—figure supplement 3A and B*). This was accompanied by an attenuation in the heart growth of MKK3/6^MCK-KO animals (*Figure 7—figure supplement 3C*), although it was not sufficient to revert it to the level of MCK-Cre control animals.

## Discussion

The present study provides several independent lines of evidence supporting a critical role for the MKK3/6–p38γ/δ signaling pathway in the development of cardiac hypertrophy. The cardiac hypertrophy developed in mice lacking MKK6 seems to be physiological in young animals, with normal or even increased cardiac function at baseline. However, aging induced the development of cardiac dysfunction and premature death. Extensive analysis with different knockout mouse models shows that in the absence of MKK6, the MKK3-stimulated p38γ/δ kinases become hyperactivated and induce enhanced postnatal hypertrophic growth through the mTOR pathway (*Figure 8*). Our results also reveal that the two main upstream p38 activators are strongly biased toward the activation of specific p38 MAPK isoforms in the heart, with p38γ/δ mainly regulated by MKK3 and p38α by MKK6, at least in homeostatic conditions.

Previous work has implicated multiple p38 family members in disease models of pathological hypertrophy, with different outcomes of p38 activation in hypertrophy and apoptosis upon different stimuli (*Nikolic et al., 2020*; *Romero-Becerra et al., 2020*), whereas p38γ and p38δ isoforms appeared to be involved primarily in regulating postnatal physiological cardiac growth and the metabolic switch during cardiac early postnatal development (*González-Terán et al., 2016*; *Santamans et al., 2021*). Such reports, however, have not addressed how different p38 MAPK isoforms are regulated, and the relative contributions of MKK3 and MKK6 to cardiac hypertrophic phenotypes have so far remained unclear. For instance, either overexpression of constitutively active or dominant negative forms of MKK3/6 in vitro result in cardiac hypertrophy (*Braz et al., 2003*; *Streicher et al., 2010*; *Zechner et al., 1997*), while in vivo transgenic overexpression of MKK6 in mouse heart did not affect cardiac hypertrophy but resulted in protection against myocardial infarction and reduced levels of markers of

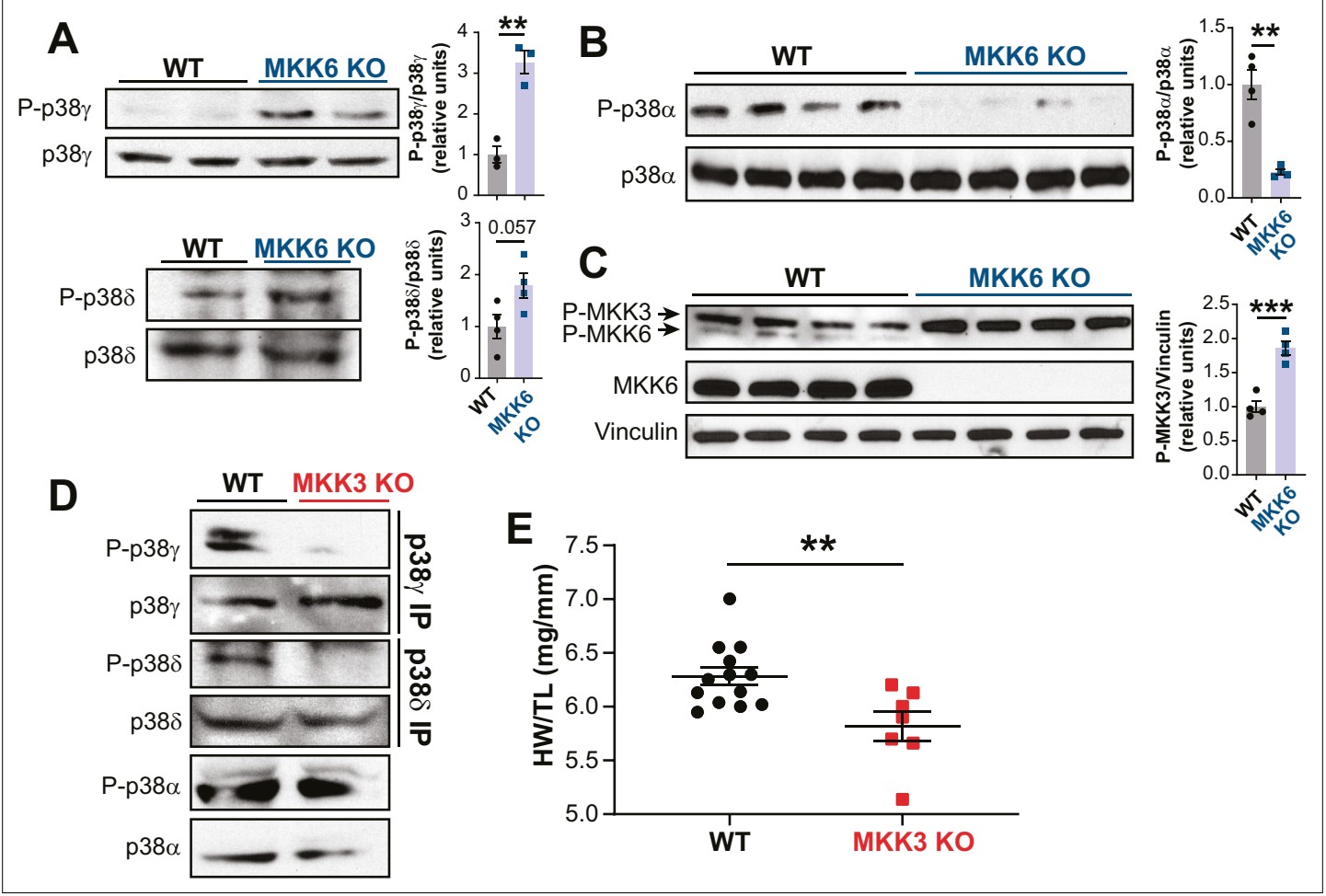

**Figure 5.** MKK6 is necessary for p38α phosphorylation and MKK3 for p38γ and p38δ phosphorylation in the heart. (**A**) Western blot analysis of the phosphorylation and amount of p38γ and δ immunoprecipitated from heart lysates from 9-week-old wild type WT (n=3–4) and MKK6 KO (n=3–4) mice. Unpaired *t*-test. (**B**) Immunoblot analysis of p38α phosphorylation and protein amount in WT (n=4) and MKK6 KO (n=4) mice. Unpaired *t*-test. (**C**) Phosphorylation and protein levels of MKK3 and MKK6 in heart lysates from WT (n=4) and MKK6 KO (n=4) mice. Unpaired *t*-test. (**D**) Immunoprecipitation and western blot analysis of the phosphorylation and protein amounts of p38α, p38γ, and p38δ isoforms in heart lysates from 9-week-old WT and MKK3 KO mice. (**E**) Heart weight to tibia length ratio in WT (n=13) and MKK3 KO (n=7) mice at 9 weeks of age. Data in E are mean ± SEM. (n=3–13). **p<0.01; ***p<0.001.

The online version of this article includes the following source data and figure supplement(s) for figure 5:

**Source data 1.** Raw blots for *Figure 5A–D*.

**Source data 2.** Raw data, statistical test and significance, and *n* for *Figure 5*, *Figure 5—figure supplement 1* and *Figure 5—figure supplement 2*.

**Figure supplement 1.** MKK3 deficiency promotes a reduced phosphorylation of p38γ/δ in the heart.

**Figure supplement 1—source data 1.** Raw blots for *Figure 5—figure supplement 1*.

**Figure supplement 2.** p38α negatively regulates MKK3 expression and phosphorylation in the heart.

**Figure supplement 2—source data 1.** Raw blots for *Figure 5—figure supplement 2*.

apoptosis (*Martindale et al., 2005*). Our results establish for the first time a complete in vivo description of the specific role of each p38 family member and their upstream kinases MKK3 and MKK6 in cardiac hypertrophy in young animals and their deleterious effects with aging. Our in vivo data support a model wherein MKK3 activity promotes hypertrophic growth. We show that MKK3 hyperactivation resulting from MKK6 deletion leads to cardiac hypertrophy, while MKK3 genetic ablation results in reduced postnatal cardiac growth. We also demonstrate that deficiency of MKK6 in cardiomyocytes is sufficient to result in cardiac hypertrophy, attributable to both, MKK6 direct function in inducing hypertrophy via p38-mediated signaling as well as its role as a negative regulator of MKK3

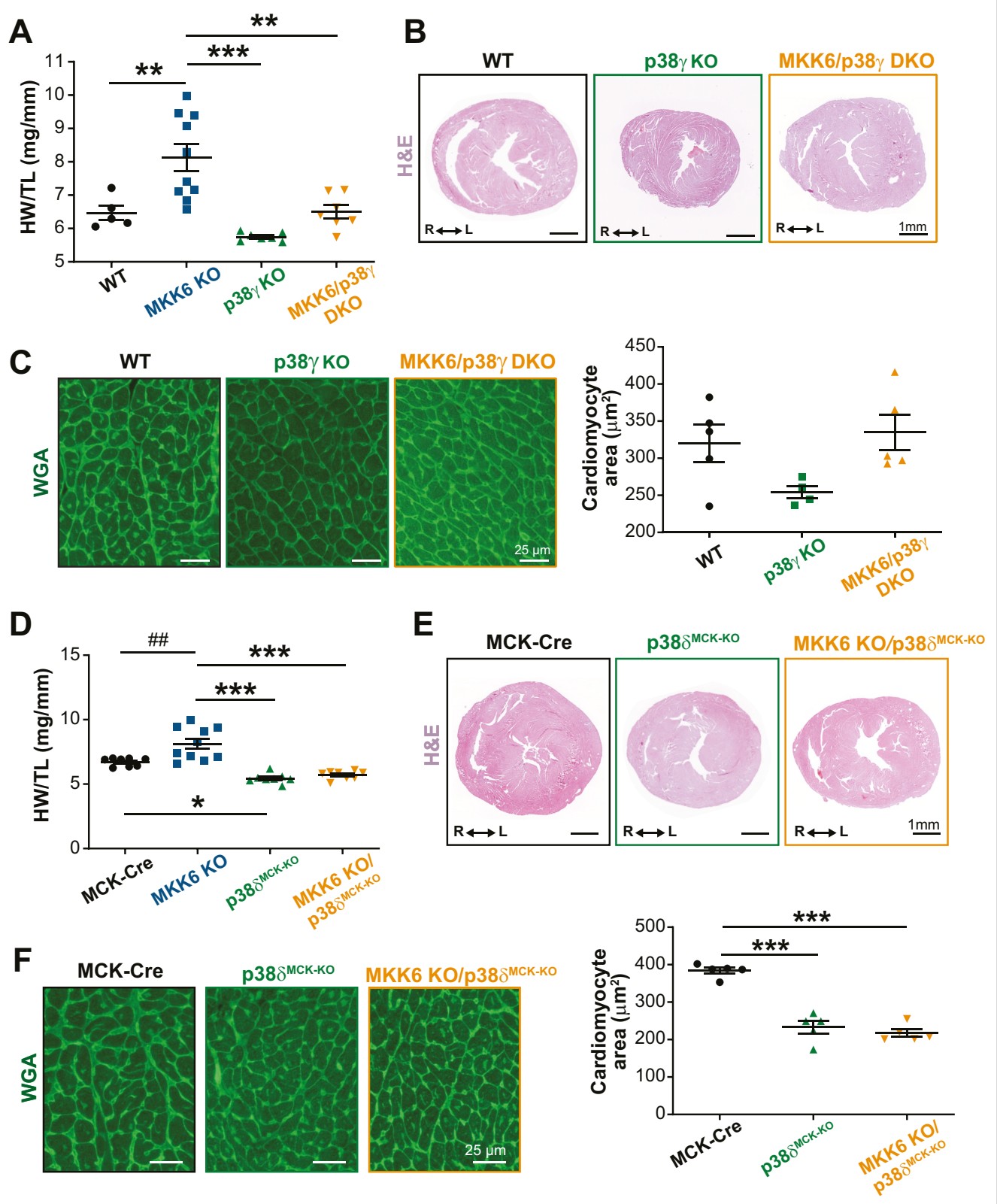

**Figure 6.** Loss of p38γ/δ in cardiomyocytes rescues the cardiac hypertrophy induced by MKK6 deficiency. All phenotypes shown come from 9-week-old mice. (**A**) Heart weight to tibia length ratio in wild type (WT) (n=5), MKK6 KO (n=10), p38γ KO (n=7), and MKK6/p38γ DKO (n=7) mice. One-way ANOVA followed by Tukey's post-test. (**B**) Representative hematoxylin and eosin (H&E)-stained transverse heart sections from WT, p38γ KO, and MKK6/p38γ DKO mice. Scale bars: 1 mm. (**C**) Representative FITC wheat germ agglutinin (FITC-WGA) staining and corresponding quantification of cardiomyocyte

*Figure 6 continued on next page*

*Figure 6 continued*

cross-sectional area in WT (n=5), p38γ KO (n=4), and MKK6/p38γ DKO (n=5) mice. Scale bars: 25 µm. One-way ANOVA followed by Tukey's post-test. (**D**) Heart weight to tibia length ratio in MCK-Cre (n=8), MKK6 KO (n=10), p38δ^MCK-KO (n=8), and MKK6 KO/p38δ^MCK-KO (n=8) mice. Kruskal-Wallis test with Dunn's post-test (##p<0.01 Mann-Whitney U test). (**E**) Representative H&E-stained transverse heart sections from MCK-Cre, p38δ^MCK-KO, and MKK6 KO/p38δ^MCK-KO mice. Scale bars: 1 mm. (**F**) Representative FITC-WGA staining and corresponding quantification of cardiomyocyte cross-sectional area in MCK-Cre (n=5), p38δ^MCK-KO (n=5), and MKK6 KO/p38δ^MCK-KO (n=5) mice. One-way ANOVA followed by Tukey's post-test. Scale bars: 25 µm. The same data from MKK6 KO mice were used in (**A**) and (**D**). Means ± SEM is shown. *p<0.05; **p<0.01; ***p<0.001; ##p<0.01.

The online version of this article includes the following source data for figure 6:

**Source data 1.** Raw data, statistical test and significance, and *n* for *Figure 6A, C and D & F*.

activity. Our results indicate that this negative regulation is mediated by p38α, both at the expression and the phosphorylation level of MKK3. This is in agreement with previous studies describing that p38α regulates the transcription of these kinases (*Ambrosino et al., 2003*). Additionally, there is evidence indicating that p38α might negatively regulate the MKK3/6 upstream kinase MAP3K TAK1 (*Singhirunnusorn et al., 2005*). Indeed, we have found that in other tissues, the lack of p38α induces p38γ and p38δ activation (*Matesanz et al., 2018*). The negative feedback activity of p38α is consistent with MKK3 hyperphosphorylation identified in MKK6 KO mice and in p38α^MCK-KO mice. Our results provide the first demonstration that MKK3 preferentially regulates p38γ and p38δ activation, whereas MKK6 is responsible for p38α regulation.

MKK6-deficient mice show p38γ/δ hyperactivation and impaired p38α activation, possibly implicating any of these isoforms in the observed phenotypes. Through the use of multiple in vivo models, we show that hyperphosphorylated MKK3 promotes cardiac hypertrophic growth through activation of p38γ/δ-DEPTOR-mediated mTOR signaling in the heart. This is consistent with our earlier work showing an essential role for these p38 isoforms in mTOR-dependent physiological and pathological cardiac hypertrophy (*González-Terán et al., 2016*), as well as with previous reports indicating that p38α does not mediate hypertrophic responses in animal models of pressure-overload cardiac hypertrophy (*Nishida et al., 2004*). This conclusion is corroborated by the reversion of cardiac hypertrophy in MKK6-deficient mice also deficient for cardiac p38γ or p38δ, and further corroboration comes from the ability of the mTOR inhibitor rapamycin to prevent cardiac hypertrophy during early postnatal cardiac development in MKK6-deficient mice. These findings correlate with impaired p38γ and p38δ activation and reduced postnatal hypertrophic growth in MKK3-deficient mice. Nevertheless, MKK3 deficiency in the context of MKK6 deletion in striated muscle produced a reduction in mTOR activation and attenuation of cardiac growth but was not able to completely rescue the normal heart size. This result suggests that, apart from mTOR activation, MKK6 deficiency might affect other signaling pathways affecting cardiac hypertrophy. For example, p38γ/δ have been implicated in the regulation of the postnatal cardiac metabolism (*Santamans et al., 2021*), which is another known direct cause of cardiac hypertrophy (*Nakamura and Sadoshima, 2018*).

The p38γ- and p38δ-mediated control of cardiac hypertrophy during postnatal cardiac development resides in cardiomyocytes (*González-Terán et al., 2016*). Accordingly, lack of MKK6 in skeletal muscle or cardiomyocytes yields the same phenotype as global MKK6 deficiency. Furthermore, the blockade of cardiac hypertrophy in MKK6-deficient mice upon deletion of p38δ specifically in striated muscle indicates that cardiac p38δ lies downstream of MKK6 in the signaling pathway controlling hypertrophic growth. Interestingly, reversion of the MKK6-deficient phenotype upon p38δ deletion was greater than that achieved upon deletion of p38γ (*Figure 6*), consistent with the reported dominance of p38δ in regulating cardiac hypertrophic growth (*González-Terán et al., 2016*).

Young MKK6-deficient mice develop a cardiac hypertrophy that could be classified as physiological, characterized by a proportionate increase in heart size with the maintenance of a normal cardiac structure. Moreover, cardiac function was normal, with increased cardiac output and stroke volume, and there was no evidence of fibrosis or re-expression of the cardiac stress fetal gene program. Physiological cardiac hypertrophy is an adaptive response that increases ventricular mass while maintaining or enhancing cardiac function (*Kang, 2006*; *Nakamura and Sadoshima, 2018*). However, it has increasingly become appreciated that sustained activation of the pathways that drive this beneficial response can ultimately result in pathological remodeling and associated sudden cardiac death (*Condorelli et al., 2002*; *Lauschke and Maisch, 2009*; *Matsui et al., 2002*; *McMullen et al., 2004a*; *Oldfield et al., 2020*; *Shioi et al., 2002*). In agreement with this, the cardiac hypertrophy observed in MKK6

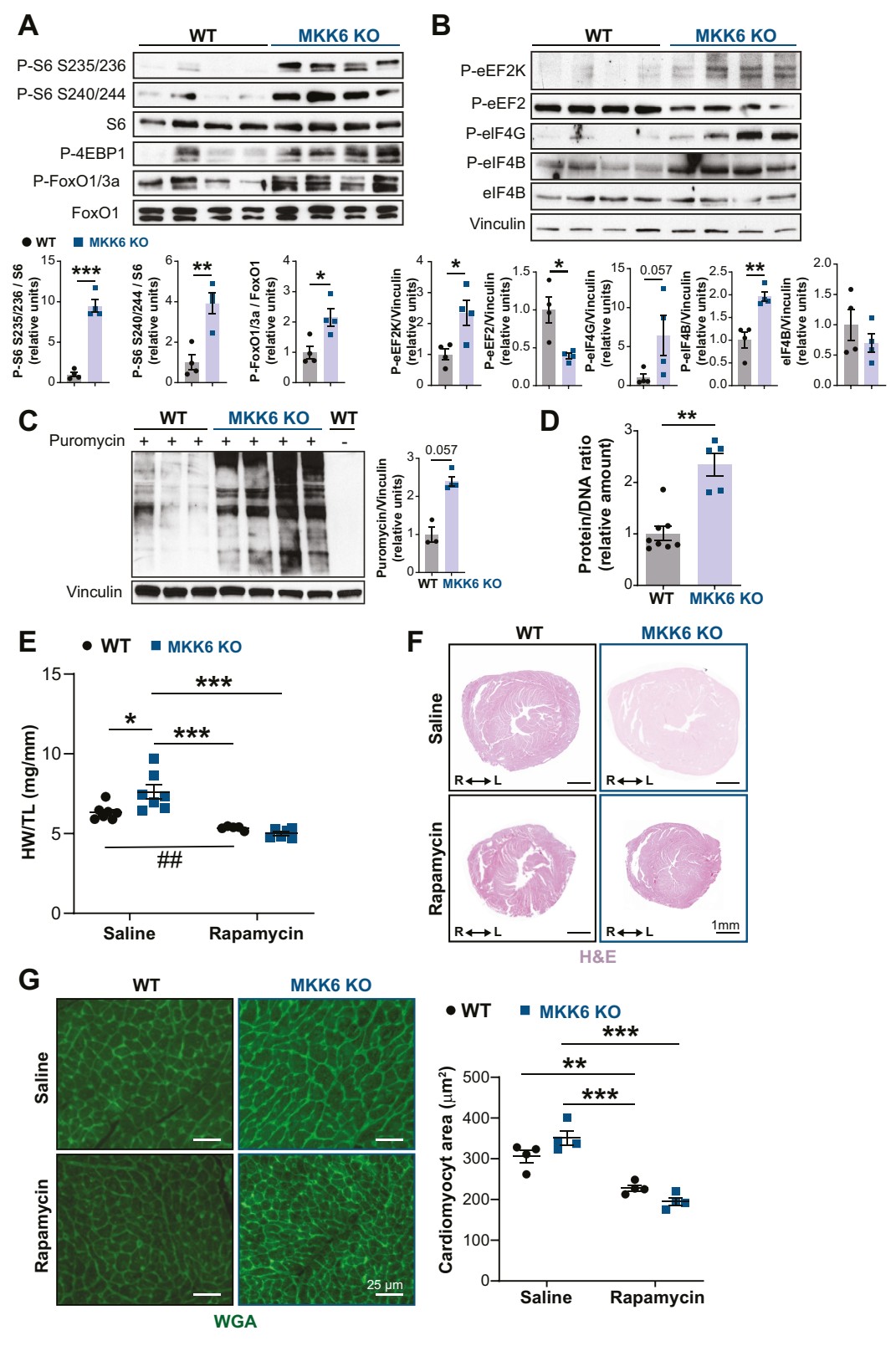

**Figure 7.** Hyperactivation of mammalian target of rapamycin (mTOR) signaling drives cardiac hypertrophy in MKK6 KO mice. (**A** and **B**) Immunoblot analysis of mTOR signaling pathway activity (**A**) and activation status of translation factors (**B**) in heart lysates from 9-week-old wild type (WT) (n=4) and MKK6 KO (n=4) mice. Unpaired *t*-test or Mann-Whitney U test. (**C**) In vivo measurement of protein synthesis in 9-week-old WT (*n=*) and MKK6 KO (n=4) mice.

*Figure 7 continued on next page*

*Figure 7 continued*

Mice were injected intraperitoneally with 0.040 µmol g$^{-1}$ puromycin dissolved in 100 µl PBS. Exactly 30 min after injection, tissues were extracted and frozen in liquid N$_2$ for immunoblot analysis with an anti-puromycin antibody. Mann-Whitney U test. (**D**) Protein content of WT (n=8) and MKK6 KO (n=5) hearts measured as the protein-DNA ratio. Mann-Whitney U test. (**E**) Heart weight to tibia length ratio in WT (n=5–7) and MKK6 KO (n=6–7) mice after rapamycin treatment. Mice received daily intraperitoneal injections with rapamycin (2 mg kg$^{-1}$ per day) or vehicle from weeks 4 to 9 after birth. Two-way ANOVA followed by Tukey's post-test (## p<0.01 unpaired *t*-test). (**F**) Representative hematoxylin and eosin (H&E)-stained transverse heart sections after treatment. Scale bars: 1 mm. (**G**) Representative FITC wheat germ agglutinin (FITC-WGA) staining and corresponding quantification of cardiomyocyte cross-sectional area from WT (n=4) and MKK6 KO (n=4) mice hearts after rapamycin treatment. Two-way ANOVA followed by Tukey's post-test. Scale bars: 25 µm. Data are mean ± SEM. *p<0.05; **p<0.01; ***p<0.001.

The online version of this article includes the following source data and figure supplement(s) for figure 7:

**Source data 1.** Raw blots for *Figure 7A–C*.

**Source data 2.** Raw data, statistical test and significance, and *n* for *Figure 7* and *Figure 7—figure supplement 1*, *Figure 7—figure supplements 3 and 4*.

**Figure supplement 1.** Mammalian target of rapamycin (mTOR)-inhibitory protein DEPTOR levels are reduced in MKK6-deficient hearts.

**Figure supplement 1—source data 1.** Raw blots for *Figure 7—figure supplement 1*.

**Figure supplement 2.** Mutant active p38α expression reverts mammalian target of rapamycin (mTOR) hyperactivation and mTOR-induced cell growth in MKK6 KO mouse embryonic fibroblasts (MEFs).

**Figure supplement 2—source data 1.** Raw blots for *Figure 7—figure supplement 2*.

**Figure supplement 3.** Tissue-specific MKK3 and MKK6 deletion in MKK3/6$^{MCK-KO}$ mice.

**Figure supplement 3—source data 1.** Raw blots for *Figure 7—figure supplement 3*.

**Figure supplement 4.** MKK3 deletion suppresses p38γ and mammalian target of rapamycin (mTOR) hyperactivation in MKK6-deficient hearts.

**Figure supplement 4—source data 1.** Raw blots for *Figure 7—figure supplement 4*.

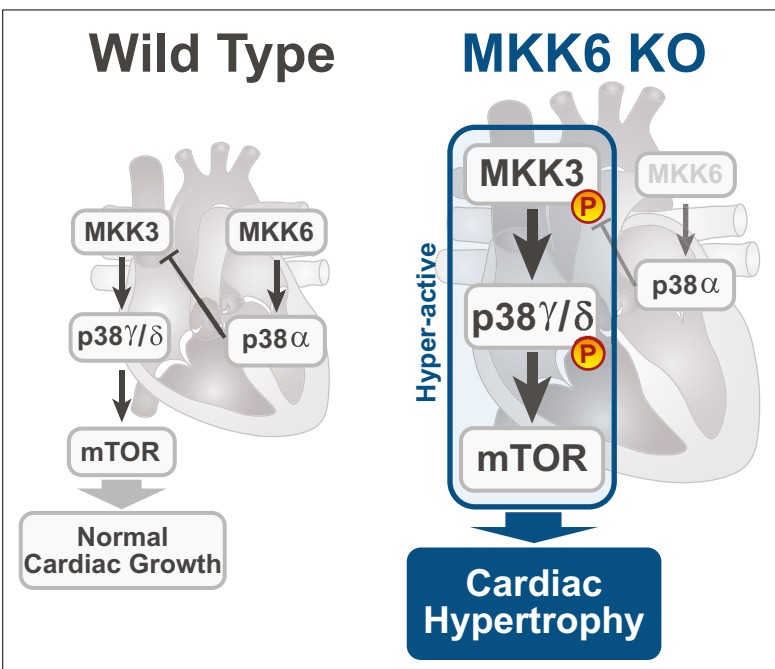

**Figure 8.** Model for p38γ/δ activation mediated cardiac hypertrophic growth. In a physiological context, MKK3-p38γ/δ pathway promotes normal cardiac growth through the activation of mammalian target of rapamycin (mTOR) signaling pathway. MKK6 deficiency stimulates the hyperactivation of MKK3-p38γ/δ and the consequent increase in mTOR activity, which drives increased cardiac hypertrophy.

KO mice becomes deleterious with age, compromising the cardiac function, and likely contributing to the reduced survival observed in these mice. We corroborated that cardiac dysfunction due to the deficiency of MKK6 restricted to cardiomyocytes in conditional KO mice also resulted in a reduced lifespan. Nevertheless, it cannot be discarded that dysfunction of other organs may also contribute to the premature death of MKK6 KO mice. Respiratory weakness related to kyphosis or ataxia could also contribute to the reduced survival of these mice at advanced ages. This would be in line with the phenotypic alterations observed in other disease mouse models with reduced lifespan, such as Duchenne muscular dystrophy or Huntington's disease, in which cardiomyopathy and heart dysfunction play an important role in disease progression, and it is the leading cause of death in patients (*Burns et al., 2015*; *Caravia et al., 2018*).

The activation of the p38α pathway has been linked to several diseases, suggesting that this pathway could represent a target for their treatment. However, the results from the clinical trials have been disappointing so far (*Canovas and Nebreda, 2021*). p38α inhibitors have been the most profoundly studied. However, these studies usually do not consider the possible undesired effect of p38α inhibition upon the other p38 pathway kinases, which could be among the reasons for failure in the outcomes. Our results show an example of how p38α inhibition leads to an unexpected activation of MKK3-p38γ/δ axis, having deleterious effects on the heart in the long term. Our finding suggests that treatment strategies using longstanding p38α inhibition should consider the potential cardiovascular risk among the possible secondary effects of the treatment. Additionally, our results indicate that inhibition of p38γ/δ could be a therapeutic strategy to treat pathologies such as hypertrophic cardiomyopathy that still remains unexplored due to the lack of specific inhibitors for these p38 family members.

## Materials and methods

### Animal preparation

MKK3 KO mice (B6.129-*Map2k3^{tm1Flv}*) (*Lu et al., 1999*; *Wysk et al., 1999*) and MKK6 KO mice (B6.129-*Map2k6^{tm1Flv}*) (*Brancho et al., 2003*) were as previously described. Mice with a germ-line mutation in the *Map2k6* or *Mapk14* gene and *LoxP* elements inserted into two introns (*Map2k6^{LoxP}* or *Mapk14^{LoxP}*) were generated as previously described (*Hui et al., 2007*; *Matesanz et al., 2017*). To generate mice lacking MKK6, p38α, p38δ, or MKK3/6 in striated muscle (MKK6^{MCK-KO}, p38α^{MCK-KO}, p38δ^{MCK-KO}, and MKK3/6^{MCK-KO}, respectively), *Map2k6^{LoxP}* or *Mapk14^{LoxP}* or *Mapk13^{LoxP}* (B6.129-Mapk13tm1) or *Map2k3^{LoxP}*/*Map2k6^{LoxP}* mice were crossed with the FVB-Tg (Ckmm-cre) 5Khn/J line (*Brüning et al., 1998*) on the C57BL/6J background (Jackson Laboratory, Strain #006405). Mice lacking MKK6 in cardiomyocytes (MKK6^{αMHC-KO}) were generated by crossing *Map2k6^{LoxP}* mice with the Tg(Myh6-cre)2182Mds line on the C57BL/6J background (Jackson Laboratory). The p38γ-negative line (B6.129-Mapk12tm1) was crossed with the MKK6 KO line (B6.129-*Map2k6^{tm1Flv}*) to generate double knockout mice (MKK6/p38γ DKO). Likewise, mice lacking p38δ in striated muscle were crossed with the MKK6 KO (B6.129-*Map2k6^{tm1Flv}*) line to generate MKK6 KO/p38δ^{MCK-KO} mice. Genotype was confirmed by PCR analysis of genomic DNA. For signaling studies, animals were killed by cervical dislocation. For rapamycin treatment, mice received daily intraperitoneal injections with rapamycin (LC Laboratories, R-5000) (2 mg kg⁻¹ per day) or vehicle (0.25% polyethylene glycol [Sigma], 0.25% Tween-20 [Sigma] in PBS); injections started at 4 weeks of age and continued until 9 weeks of age, when heart size was analyzed by echocardiography. All animal procedures conformed to EU Directive 86/609/EEC and Recommendation 2007/526/EC regarding the protection of animals used for experimental and other scientific purposes, enacted under Spanish law 1201/2005.

### Computed tomography scan

CT studies were performed with a small-animal PET/CT scanner (nanoScan, Mediso, Hungary). For the acquisition, mice were anesthetized using isoflurane 2% and 1.8 l/min oxygen flow. Ophthalmic gel was placed in the eyes to prevent drying. CT was acquired using an X-ray beam current of 178 μA and a tube voltage of 55 kVp with 360 projections of 500 ms in a helical scan with pitch 1 and binning 1:4. CT image was reconstructed using a Ramlack algorithm with a final resolution of 0.078 mm³.

## Histology

Tissue samples were fixed in 10% formalin for 48 hr, dehydrated, and embedded in paraffin. Sections (8 µm) were cut and stained with hematoxylin and eosin (American Master Tech Scientific). Fibrosis was assessed by picrosirius red staining (Sigma), and the positive area for fibrosis was quantified with Image J software (*Schneider et al., 2012*). For wheat germ agglutinin (WGA) immunofluorescence, 8 µm heart sections were prepared, washed in PBS, incubated for 2 hr in WGA-Alexa 488 lectin (Invitrogen, Carlsbad, CA, USA), and washed and mounted in anti-fade reagent. Four images (×20) were taken from each heart, and the areas of 100–200 cross-sectionally oriented cardiomyocytes were measured and analyzed with Image J software (*Schneider et al., 2012*).

## Echocardiography

Mice were anesthetized by inhalation of isoflurane and oxygen (1.25 and 98.75%, respectively), and echocardiography was performed with a 30-MHz transthoracic echocardiography probe. Images were obtained with the Vevo 2100 micro-ultrasound imaging system (VisualSonics, Toronto, Canada). Short-axis, long-axis, B-mode, and two-dimensional M-mode views were obtained. Scans were conducted by two experienced researchers blinded to the mouse genotype. Measurements of left parasternal long and short axes and M-mode images (left parasternal short-axis) were obtained at a heart rate of 500–550 beats per minute. LV end-diastolic diameter (LVEDD), LV end-systolic diameter (LVESD), and wall thickness were measured from M-mode tracings, and the average of three consecutive cardiac cycles is reported. The LV fractional shortening percentage was calculated as ([LVEDD−LVESD]/LVEDD)×100. MRI of lung was performed with a 7T Agilent scanner (Agilent, Santa Clara, CA, USA) equipped with a DD2 console and an actively shielded gradient set (205/120 insert of maximum 130 mT m$^{-1}$ gradient strength). To enhance the signal-to-noise ratio during image acquisition, we used a combination of a 72 mm inner diameter quadrature birdcage TX volume coil (Rapid Biomedical GmBH, Germany) and an actively detuning 30 mm flexible customized surface RX coil (Neos Biotec, Pamplona, Spain). After acquisition of a tripilot gradient-echo image, a gradient-echo sequence without gating was used to acquire oblique coronal slices (1–2 slices) and axial slices (7–10 slices) covering the entire lung, 72 s acquisition time per slice) using the following parameters: TR/TE = 6.7/2.2 ms, flip angle = 10°, bandwidth = 100 kHz, field of view = 3 × 3 cm, matrix = 256 × 128, and slice thickness = 1 mm (ref. 40). These images were used to determine interventricular septum and LV posterior wall thicknesses and LV corrected mass; the short-axis M-mode quantification was chosen as the most representative. The cardiac function was estimated from the ejection fraction and fractional shortening obtained from M-mode views by a blinded echocardiography expert. For ejection fraction measurements, a long- or short-axis view of the heart was selected to obtain an M-mode registration in a line perpendicular to the left ventricular septum and posterior wall at the level of the mitral chordae tendineae.

## Immunoblot analysis

Tissue extracts were prepared in Triton lysis buffer (20 mM Tris [pH 7.4], 1% Triton X-100, 10% glycerol, 137 mM NaCl, 2 mM EDTA, 25 mM β-glycerophosphate, 1 mM sodium orthovanadate, 1 mM phenylmethylsulfonyl fluoride, and 10 µg ml$^{-1}$ aprotinin and leupeptin). Extracts (20–50 µg protein) and immunoprecipitates (prepared from 0.5 to 3 mg) were examined by immunoblot. For immunoprecipitation assays, heart extracts were incubated with 1–5 µg of a specific antibody coupled to protein-G-Sepharose. After incubation overnight at 4°C with agitation, the captured proteins were centrifuged at 10,000 g, the supernatants collected, and the beads washed four times in PBS1X. Beads were boiled for 5 min at 95°C in 10 µl sample buffer. Extracts and immunoprecipitates were examined by SDS–PAGE and blotted with antibodies to the following targets: p38γ and p38δ (*Sabio et al., 2005*; *Sabio et al., 2004*) at 1 µg ml$^{-1}$; vinculin (Sigma); puromycin (Millipore clone 12D10); phospho-MKK3 (Ser189)/MKK6 (Ser207) (#9231S), MKK3 (#9232S), MKK6 (#9264S), phospho-p38 MAPK (Thr180/Tyr182) (#9211S), phospho-mTOR (Ser2481) (#2974S), mTOR (#2972), phospho-p70S6 kinase (Thr 389) (108D2) (#9234S), p70S6 kinase (#9202S), phospho-S6 (Ser 235/236) (D57.2.2E) (#4858S), phospho-S6 (Ser 240/244) (61H9) (#4838S), S6 ribosomal protein (5G10) (#2217S), phospho-FoxO1 (Thr24)/FoxO3a (Thr32) (#9464S), Fox01 (C29H4) (#2880S), phospho-eEF2 (Thr56) (#2331S), phosphor-eIF4G (Ser1108) (#2441S), phospho-eIF4B (Ser422) (#6591S), eIF4B (#3592S), phospho-4EBP1 (Thr37/46) (#9459S), 4EBP1 (#9452), and DEPTOR (D9F5) (#11,816) (Cell Signaling Technology); phospho-p38

MAPK gamma/delta (Tyr185, Tyr182) (#PA5-105907, ThermoFisher Scientific); p38α (C-20) (#sc535-G, Santa Cruz Biotechnology), all at a 1:1000 dilution. In some cases, the membranes were stripped, or if the primary antibodies are raised in different species, they were treated with sodium azide. Immunocomplexes were detected by enhanced chemiluminescence (GE Healthcare Lifesciences). Immunoblots were quantified using Image J software (*Schneider et al., 2012*).

### Cell lines and cell culture

Immortalized MEFs from WT and MKK6 KO mice were cultured in Dulbecco's Modified Eagle Medium (DMEM) supplemented with 10% heat-inactivated fetal bovine serum (Gibco, #1027–106), glutamine 2 mM (Hyclone, #SH30034.01), and penicillin/streptomycin 100 µg ml$^{-1}$ (Sigma, #P4333). Cells were grown at 37°C in a 5% $CO_2$ humid atmosphere. Cells were tested to confirm absence of *Mycoplasma* contamination (MycoAlert PLUS Mycoplasma Detection Kit, Lonza). Cells were infected with 3.83E06 to 4.67E06 transducing units of blank control (abm, #LV590) or p38α active (D176A/F327S) (VectorBuilder, VB170531-1024adk) lentivirus (second generation, VSV-G pseudotype). Cell area was quantified using Image J software (*Schneider et al., 2012*) and Cellpose algorithm (*Stringer et al., 2021*).

### In vivo protein synthesis assay

For all in vivo measurements of protein synthesis, mice were injected intraperitoneally with 0.040 µmol g$^{-1}$ puromycin dissolved in 100 µl PBS. Exactly 30 min after injection, tissues were extracted and frozen in liquid $N_2$ for subsequent immunoblot analysis of protein-incorporated puromycin.

### Blood pressure and heart rate measurements

Blood pressure and heart rate in mice were measured using the non-invasive tail-cuff method (*Kubota et al., 2006*). The measures were performed in conscious mice placed in a BP-2000 Blood Pressure Analysis System (Visitech Systems). 10 preliminary measurements and 10 actual measurements were recorded, and the average of the 10 actual measurements was used for analysis. The animals were trained for four consecutive days prior to the actual measurements were registered. All the measurements were taken at the same time of the day.

### Reverse transcription quantitative real-time polymerase chain reaction

RNA 500 ng – extracted with RNAeasy Plus Mini kit (Qiagen) following manufacturer instructions – was transcribed to cDNA, and RT-qPCR was performed using Fast Sybr Green probe (Applied Biosystems) and the appropriate primers in the 7900 Fast Real Time thermocycler (Applied Biosystems). Relative mRNA expression was normalized to *Gapdh* mRNA measured in each sample. The following primer sequences were used: *Fn1* Fw: ATGTGGACCCCTCCTGATAGT,Rev: GCCCAGTGATTTCAGCAAAG G; Col1a1 Fw: GCTCCTCTTAGGGGCCACT,Rev: CCACGTCTCACCATTGGGG; Col3a1 Fw: CTGT AACATGGAAACTGGGGAAA,Rev: CCATAGCTGAACTGAAAACCACC; Nppa Fw: GCTTCCAGGCCA TATTGGAG,Rev: GGGGGCATGACCTCATCTT; Nppb Fw: GAGGTCACTCCTATCCTCTGG,Rev: GCCA TTTCCTCCGACTTTTCTC; Acta-2 Fw: CCCAAAGCTAACCGGGAGAAG, Rev: CCAGAATCCAAC ACGATGCC; Myh7 Fw: ACTGTCAACACTAAGAGGGTCA,Rev: TTGGATGATTTGATCTTCCAGGG; Gapdh Fw: TGAAGCAGGCATCTGAGGG,,Rev: CGAAGGTGGAAGAGTGGGA.

### Statistical analysis

Results are expressed as mean ± SEM. A difference of $p < 0.05$ was considered significant. Gaussian (normal) distribution was determined using the Shapiro-Wilks normality test. For normally distributed populations, differences between groups were examined for statistical significance by two-tailed Student *t*-test (two groups) and one-way ANOVA followed by Tukey post-test (three or more groups). To test the respective roles of treatment or age and genotype, a two-way ANOVA was performed. Tukey or Sidak post-test was subsequently employed when appropriate. For data that failed normality testing, Mann-Whitney U test (two groups) or Kruskal-Wallis with Dunn post-test (three or more groups) was performed. Gehan-Breslow-Wilcoxon test was used to assess significance in the Kaplan–Meier survival analysis.

## Acknowledgements

We thank S Bartlett and F Chanut for English editing. We are grateful to RJ Davis, A Padmanabhan, M Costa and C López-Otín for critical reading of the manuscript. We thank Dr. RJ Davis for the MKK3 and MKK6 KO animals and Dr. Erwin F Wagner for the p38α flox mice. We thank AC Silva (ana@anasilva illustrations.com) for help with figure editing and design. This work was funded by a CNIC Intramural Project Severo Ochoa (Expediente 12–2016 IGP) to GS and JJ and PID2019-104399RB-I00 funded by MCIN/AEI/10.13039/501100011033 to GS. BGT was a fellow of FPI Severo Ochoa CNIC Program (SVP-2013-067639) and is an American Heart Association Postdoctoral Fellow (18POST34080175). RRB is a fellow of the FPU Program (FPU17/03847). The following grants provided additional funding: GS is granted by funds from European Regional Development Fund (ERDF): EFSD/Lilly European Diabetes Research Programme Dr Sabio, Fundación AECC PROYE19047SABI and Comunidad de Madrid IMMUNOTHERCAN-CM B2017/BMD-3733; US National Heart, Lung, and Blood Institute (R01 Grant HL122352), Fondos FEDER, Madrid, Spain, and Fundación Bancaria "La Caixa (project HR19/52160013); Fundación La Marató TV3: Ayudas a la investigación en enfermedades raras 2020 (LA MARATO-2020); and Instituto de Salud Carlos III to JJ. IN was funded by EFSD/Lilly grants (2017 and 2019), the CNIC IPP FP7 Marie Curie Programme (PCOFUND-2012–600396), EFSD Rising Star award (2019), JDC-2018-Incorporación (MIN/JDC1802). The CNIC is supported by the Instituto de Salud Carlos III (ISCIII), the Ministerio de Ciencia e Innovación (MCIN) and the Pro CNIC Foundation

## Additional information

### Funding

| Funder | Grant reference number | Author |
| --- | --- | --- |
| Centro Nacional de Investigaciones Cardiovasculares | CNIC Intramural Project Severo Ochoa (Expediente 12–2016 IGP) | Guadalupe Sabio José Jalife |
| Ministerio de Ciencia e Innovación/Agencia Estatal de Investigación | PID2019-104399RB-I00 funded by MCIN/AEI/10.13039/501100011033 | Guadalupe Sabio |
| FPI Severo Ochoa CNIC Program | SVP-2013-067639 | Barbara Gonzalez-Teran |
| American Heart Association | 18POST34080175 | Barbara Gonzalez-Teran |
| Ministerio de Ciencia, Innovación y Universidades | FPU17/03847 | Rafael Romero-Becerra |
| European Foundation for the Study of Diabetes | EFSD/Lilly European Diabetes Research Programme Dr Sabio | Guadalupe Sabio |
| Fundación Científica Asociación Española Contra el Cáncer | PROYE19047SABI | Guadalupe Sabio |
| Comunidad de Madrid | IMMUNOTHERCAN-CM B2017/BMD-3733 | Guadalupe Sabio |
| National Heart, Lung, and Blood Institute | R01 HL122352 | José Jalife |
| Fondo Europeo de Desarrollo Regional | Madrid, Spain | José Jalife |
| Fundación Bancaria "La Caixa" | HR19/52160013 | José Jalife |
| Fundación La Marató TV3 | Ayudas a la investigación en enfermedades raras 2020 (LA MARATO-2020) | José Jalife |
| Instituto de Salud Carlos III | | José Jalife |

| Funder | Grant reference number | Author |
|---|---|---|
| European Foundation for the Study of Diabetes | EFSD/Lilly 2017 | Ivana Nikolic |
| European Foundation for the Study of Diabetes | EFSD/Lilly 2019 | Ivana Nikolic |
| FP7 People: Marie Skłodowska-Curie Actions | PCOFUND-2012-600396 | Ivana Nikolic |
| European Foundation for the Study of Diabetes | Rising Star award (2019) | Ivana Nikolic |
| JDC-2018-Incorporación | MIN/JDC1802 | Ivana Nikolic |
| American Heart Association | 818798 | Barbara Gonzalez-Teran |

The funders had no role in study design, data collection and interpretation, or the decision to submit the work for publication.

## Author contributions

Rafael Romero-Becerra, Barbara Gonzalez-Teran, Data curation, Formal analysis, Validation, Investigation, Visualization, Methodology, Writing – original draft, Writing – review and editing; Alfonso Mora, Data curation, Investigation, Visualization, Methodology, Writing – review and editing; Elisa Manieri, Ivana Nikolic, Ayelén Melina Santamans, Valle Montalvo-Romeral, Elena Rodríguez, Luis Leiva-Vega, Laura Sanz, Víctor Bondía, Luis Jesús Jiménez-Borreguero, Investigation, Methodology; Francisco Miguel Cruz, David Filgueiras-Rama, José Jalife, Investigation, Methodology, Writing – review and editing; Marta León, Investigation; Guadalupe Sabio, Conceptualization, Resources, Formal analysis, Supervision, Funding acquisition, Investigation, Visualization, Methodology, Writing – original draft, Project administration, Writing – review and editing

## Author ORCIDs

Rafael Romero-Becerra http://orcid.org/0000-0003-3935-2647
Alfonso Mora http://orcid.org/0000-0002-6397-4836
Luis Jesús Jiménez-Borreguero http://orcid.org/0000-0001-5870-237X
José Jalife http://orcid.org/0000-0003-0080-3500
Barbara Gonzalez-Teran http://orcid.org/0000-0002-4336-8644
Guadalupe Sabio http://orcid.org/0000-0002-2822-0625

## Ethics

This study was performed in strict accordance with the recommendations in the Guide for the Care and Use of Laboratory Animals of the National Institutes of Health. All animal procedures conformed to EU Directive 86/609/EEC and Recommendation 2007/526/EC regarding the protection of animals used for experimental and other scientific purposes, enacted under Spanish law 1201/2005. All of the animals were handled according to approved institutional animal care and use committee protocols (PROEX-215/18) of the Comunidad de Madrid.

## Decision letter and Author response

Decision letter https://doi.org/10.7554/eLife.75250.sa1
Author response https://doi.org/10.7554/eLife.75250.sa2

# Additional files

## Supplementary files

• Transparent reporting form

## Data availability

All data generated or analyzed during this study are included in the manuscript and supporting files; source data files have been provided for Figures 1–7.

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
