## [Editor Report]

The paper for the first time shows that MKK6 reduces life span in mice, and that young mice with MKK deficiency display cardiac hypertrophy and that it progresses to cardiac dilatation and fibrosis as they age. The paper also provides a mechanism for this phenomenon and demonstrate that MKK6 deletion leads to reduced p38a activation but at the same time causes MKK3-p38g/d hyperphosphorylation and increased mTOR signaling. These studies are novel and will advance our understanding of the role of this pathway in aging.

---

## [Decision Letter]

**Decision letter after peer review:**

Thank you for submitting your article "MKK6 deficiency promotes cardiac dysfunction through MKK3-p38γ/δ-mTOR hyperactivation" for consideration by *eLife*. Your article has been reviewed by 3 peer reviewers, one of whom is a member of our Board of Reviewing Editors, and the evaluation has been overseen by a Reviewing Editor and Mone Zaidi as the Senior Editor. The reviewers have opted to remain anonymous.

Essential revisions:

We believe it is important to address the comments of all reviewers, specifically the following:

1. The authors recently published a review paper in International Journal of Molecular Sciences on the role of P38 MAPK pathway in the heart. It is important for the authors to better highlight the significance of their findings over what has been published in the past.

2. In figure 5, more immunoblots demonstrating reduced phosphorylation of p38g/d are needed. Since this is a major part of the conclusion, the authors need to report more than N=1.

3. In figure 7, P-eEF2K blot is not publication quality and should be repeated.

4. The experiments in Figure 7E do not make sense. Are the authors implying that rapamycin has no effect on cardiac hypertrophy in WT mice, but reduces cardiac hypertrophy in MKK6 KO mice? This interpretation is incorrect, since mTOR has an effect in cardiac hypertrophy even in WT mice. Additionally, it is hard to interpret the data in that figure, since significance between saline and rapamycin in WT mice is not provided. Finally, the authors noted a difference in WT in cardiomyocyte area in Figure 7G.

5. Although this claim is made in the paper, the authors fail to provide evidence that p38a inhibits MKK3.

6. p38 can be activated by AMPK in certain contexts, and the authors have previously shown that Mkk6 deficiency can activates AMPK. What is the status of the cardiac AMPK pathway in striated and cardiac-specific Mkk6 knockout mice? If AMPK is inhibited, is there any change in p38 status?

7. The ataxic phenotype of the Mkk6 knockout mice is striking, however the authors do not elaborate on this. How does this fit in with the age-related cardiac dysfunction? Are these separate pathologic processes or part of a unifying syndrome? Do the authors believe these mice die prematurely due to ataxia and respiratory weakness related to kyphosis, or the cardiac dysfunction, or both?

8. MKK3 data are associative. The authors should test whether MKK3 inhibition is able to attenuate p38γ/δ and mTOR activation and attenuate cardiac growth in MKK6 KO mice. Similarly, it would be important to evaluate whether cardiac overexpression of p38α is able to rescue cardiac hypertrophy and mTOR activation in MKK6 KO mice. In case the authors cannot perform these experiments, similar studies should at least be performed in primary cardiomyocytes in vitro.

9. The authors should check whether MKK3 interaction with p38γ/δ is enhanced in MKK6 KO mice. Proximity ligation assay would be useful for these experiments, also to get additional insights into the subcellular compartments in which these interactions may occur.

10. p38 α/γ/δ activity should be measured in MKK6 KO mice.

11. In the proposed mechanism (Figure 8), the authors suggest that p38α inhibits MKK3 in wild type mice. It is not clear how p38α regulates MKK3. Please address this issue.

12. The cause of death in aged systemic MKK6 KO mice should be understood. It is not clear whether these mice die because of cardiac abnormalities or through other non-cardiac issues. Do cardiomyocyte-specific MKK6 mice develop cardiac dysfunction and die prematurely?

13. The authors claim that activation of mTOR in MKK6 -/- mice is a direct consequence of p38γ/δ activation. However, this statement should be supported by additional evidence, also clarifying how p38γ/δ would regulate mTOR. DEPTOR phosphorylation and expression should be analysed, as performed in previous work from the same group (PMID: 26795633).

*Reviewer #1 (Recommendations for the authors):*This is a well written paper and the data support the conclusion. As mentioned above, the novelty of the paper remains a major concern for this reviewer.

1. The authors recently published a review paper in International Journal of Molecular Sciences on the role of P38 MAPK pathway in the heart. It is important for the authors to better highlight the significance of their findings over what has been published in the past.

2. In figure 5, more immunoblots demonstrating reduced phosphorylation of p38g/d are needed. Since this is a major part of the conclusion, the authors need to report more than N=1.

3. In figure 7, P-eEF2K blot is not publication quality and should be repeated.

4. The experiments in Figure 7E do not make sense to this reviewer. Are the authors implying that rapamycin has no effect on cardiac hypertrophy in WT mice, but reduces cardiac hypertrophy in MKK6 KO mice? This interpretation is incorrect, since mTOR has an effect in cardiac hypertrophy even in WT mice. Additionally, it is hard to interpret the data in that figure, since significance between saline and rapamycin in WT mice is not provided. Finally, the authors noted a difference in WT in cardiomyocyte area in Figure 7G.

5. Although this claim is made in the paper, the authors fail to provide evidence that p38a inhibits MKK3.

*Reviewer #2 (Recommendations for the authors):*

This is an important and well-written manuscript. However, there are also several concerns as described below.

1. p38 can be activated by AMPK in certain contexts, and the authors have previously shown that Mkk6 deficiency can activates AMPK. What is the status of the cardiac AMPK pathway in striated and cardiac-specific Mkk6 knockout mice? If AMPK is inhibited, is there any change in p38 status?

2. The ataxic phenotype of the Mkk6 knockout mice is striking, however the authors do not elaborate on this. How does this fit in with the age-related cardiac dysfunction? Are these separate pathologic processes or part of a unifying syndrome? Do the authors believe these mice die prematurely due to ataxia and respiratory weakness related to kyphosis, or the cardiac dysfunction, or both?

*Reviewer #3 (Recommendations for the authors):*

My specific comments for the authors follow:

1) MKK3 data are associative. The authors should test whether MKK3 inhibition is able to attenuate p38γ/δ and mTOR activation and attenuate cardiac growth in MKK6 KO mice. Similarly, it would be important to evaluate whether cardiac overexpression of p38α is able to rescue cardiac hypertrophy and mTOR activation in MKK6 KO mice. In case the authors cannot perform these experiments, similar studies should at least be performed in primary cardiomyocytes in vitro.

2) The authors should check whether MKK3 interaction with p38γ/δ is enhanced in MKK6 KO mice. Proximity ligation assay would be useful for these experiments, also to get additional insights into the subcellular compartments in which these interactions may occur.

3) Is MKK3 overexpression alone sufficient to induce cardiomyocyte hypertrophy in vitro through p38γ/δ-dependent mechanisms?

4) p38 α/γ/δ activity should be measured in MKK6 KO mice.

5) In the proposed mechanism (Figure 8), the authors suggest that p38α inhibits MKK3 in wild type mice. It is not clear how p38α regulates MKK3. Please address this issue.

6) The cause of death in aged systemic MKK6 KO mice should be understood. It is not clear whether these mice die because of cardiac abnormalities or through other non-cardiac issues. Do cardiomyocyte-specific MKK6 mice develop cardiac dysfunction and die prematurely?

7) It would be interesting to test whether p38γ/δ inhibition reduces cardiac dysfunction and mortality in aged MKK6 KO mice. Please address this issue.

8) The authors claim that activation of mTOR in MKK6 -/- mice is a direct consequence of p38γ/δ activation. However, this statement should be supported by additional evidence, also clarifying how p38γ/δ would regulate mTOR. DEPTOR phosphorylation and expression should be analysed, as performed in previous work from the same group (PMID: 26795633).

9) Experiments performed with rapamycin should be implemented. The authors should check whether mTOR was effectively inhibited after rapamycin treatment, along with protein synthesis. Furthermore, the effects of mTOR inhibition should also be evaluated in aged MKK6 -/- mice.

10) Figure 7A-B. p-4EBP1 and p-eIF4G levels should be normalized for total p-4EBP1 and p-eIF4G, respectively, as performed for the other markers.

11) Representative echocardiographic images should be presented.

12) In Figures 6B, C, E, F, a control MKK6 -/- group is missing.

[Editors' note: further revisions were suggested prior to acceptance, as described below.]

Thank you for resubmitting your work entitled "MKK6 deficiency promotes cardiac dysfunction through MKK3-p38γ/δ-mTOR hyperactivation" for further consideration by *eLife*. Your revised article has been evaluated by Mone Zaidi (Senior Editor) and a Reviewing Editor.

The manuscript has been improved but there are some remaining issues that need to be addressed, as outlined below:

*Reviewer #2 (Recommendations for the authors):*

The authors have adequately addressed all reviewer comments and the manuscript is now suitable for publication

*Reviewer #3 (Recommendations for the authors):*

The authors made significant efforts to address my criticisms and the manuscript is significantly improved. However, some concerns remain and several data are still associative. In addition, important findings obtained during revision were not included in the actual version of the manuscript. In details:

1) Figure 7 (rebuttal letter). Since MKK3 deletion does not rescue cardiac hypertrophy in MKK6 -/- mice, additional molecular players may be involved in addition to MKK3/mTOR signaling. Please discuss this aspect. Otherwise, the authors should evaluate cardiac hypertrophy as performed in Figure 2 and not only by the evaluation of heart weight.

2) I understand the difficulties in generating mice with overexpression of p38α. However, in order to better dissect the proposed molecular mechanisms, the authors should test at least in vitro whether p38α overexpression decreases mTOR activation and cell hypertrophy in the presence of MKK6 deletion.

3) The authors should discuss the new findings in the text.

4) Western blots in the manuscript should be quantified.

---

## [Author Response]

Essential revisions:We believe it is important to address the comments of all reviewers, specifically the following:1. The authors recently published a review paper in International Journal of Molecular Sciences on the role of P38 MAPK pathway in the heart. It is important for the authors to better highlight the significance of their findings over what has been published in the past.

We apologize for not including a more extensive review of the current knowledge of the role of p38 pathway in heart function to highlight the novelty of our results. We have now included this view in the manuscript and highlight the significance of our findings.

2. In figure 5, more immunoblots demonstrating reduced phosphorylation of p38g/d are needed. Since this is a major part of the conclusion, the authors need to report more than N=1.

We apologize for having included only one of the representative Immunoprecipitation blots. Now we have included the replicates in Figure 1 of this letter and Figure 5 —figure supplement 1A.

3. In figure 7, P-eEF2K blot is not publication quality and should be repeated.

We apologize for the low-quality blot. Sometimes performing immunoblots from animal cardiac tissue in homeostatic condition is challenging. We have now included a new blot with better quality (Figure 7B).

4. The experiments in Figure 7E do not make sense. Are the authors implying that rapamycin has no effect on cardiac hypertrophy in WT mice, but reduces cardiac hypertrophy in MKK6 KO mice? This interpretation is incorrect, since mTOR has an effect in cardiac hypertrophy even in WT mice. Additionally, it is hard to interpret the data in that figure, since significance between saline and rapamycin in WT mice is not provided. Finally, the authors noted a difference in WT in cardiomyocyte area in Figure 7G.

We thank the reviewer for the comment and we are sorry to have not explained properly these results. Rapamycin treatment had, indeed, an effect on cardiac hypertrophy in WT mice. However, this was not strong enough to be significant in a Tukey’s post hoc test following the 2-way ANOVA test that corrects for the multiple comparisons. However, the comparison of the heart weight to tibia length ratio to compare WT saline vs WT rapamycin with an unpaired t-test (or with a Fisher’s Least Significant Difference (LSD) test following the 2-way ANOVA, p=0.0257) shows that rapamycin administration significantly reduced the heart growth in these animals (Figure 7E). We have included now the significance of this comparison in Figure 7 and in the source data for Figure 7.

5. Although this claim is made in the paper, the authors fail to provide evidence that p38a inhibits MKK3.

We thank the reviewer for this excellent comment. To address this question, we have generated the conditional p38a KO mouse in striated muscle (p38a^MCK-KO^) to evaluate this important point. We found that lack of p38a results in increased activation of MKK3 and MKK6 in the heart, accompanied by higher levels of these proteins. These results are in agreement with previous studies describing that p38a can control their transcription (Ambrosino et al., 2003). Our results indicate that p38a regulates MKK3 in the heart by controlling its expression and activation. We have included these results in Figure 5 —figure supplement 2A.

6. p38 can be activated by AMPK in certain contexts, and the authors have previously shown that Mkk6 deficiency can activates AMPK. What is the status of the cardiac AMPK pathway in striated and cardiac-specific Mkk6 knockout mice? If AMPK is inhibited, is there any change in p38 status?

We agree with the reviewer that this is an interesting question. We have checked AMPK phosphorylation in heart from MKK6 KO mice to evaluate its activation and we did not find differences between WT and MKK6 deficient mice (Author response image 1). Our previous work shows that MKK6 deficiency can activate AMPK in adipose tissue by modulation of T3 response. The different outcomes in AMPK activation resulting from MKK6 deficiency between tissues is not surprising given the different involvement of the signaling in each tissue.

**Author response image 1. sa2fig1:** AMPK phosphorylation is not altered in MKK6-deficient hearts. Immunoblot analysis of AMPKa phosphorylation and protein levels in hearts from 9-week-old WT and *Mkk6^-/-^* mice.

7. The ataxic phenotype of the Mkk6 knockout mice is striking, however the authors do not elaborate on this. How does this fit in with the age-related cardiac dysfunction? Are these separate pathologic processes or part of a unifying syndrome? Do the authors believe these mice die prematurely due to ataxia and respiratory weakness related to kyphosis, or the cardiac dysfunction, or both?

We agree that this is an important question, which is the cause of the age-related death of MKK6 KO mice. To be able to answer it, we have aged conditional KO mice lacking MKK6 specifically in cardyomyocytes (Mkk6^aMHC-KO^). We have found that deletion of MKK6 restricted to heart is enough to induce premature death (Figure 4—figure supplement 2A). However, we agree with the reviewer that we cannot discard that, in the whole-body KO animals, dysfunction of other tissues may also contribute to the death of the mice. We have addressed this issue in the discussion.

8. MKK3 data are associative. The authors should test whether MKK3 inhibition is able to attenuate p38γ/δ and mTOR activation and attenuate cardiac growth in MKK6 KO mice. Similarly, it would be important to evaluate whether cardiac overexpression of p38α is able to rescue cardiac hypertrophy and mTOR activation in MKK6 KO mice. In case the authors cannot perform these experiments, similar studies should at least be performed in primary cardiomyocytes in vitro.

We appreciate the excellent comment of the reviewer. To answer this question, we have now generated the double conditional mice lacking both, MKK6 and MKK3, as there are not good specific inhibitor for MKK3 protein and the whole-body double KO animals die during embryonic development due to defects in placenta formation and deficiencies in the development of embryonic vasculature (Brancho et al., 2003). We have found that indeed MKK3 deletion is able to reduced p38γ and mTOR activation and attenuate cardiac growth in muscle-specific MKK6 KO mice (Author response image 2).

We have had problems overexpressing p38a in animals lacking MKK6. The reason is that to overexpress proteins in the heart we inject offspring at P1 and MKK6 mothers are very nervous and usually kill the offspring. In addition, the animals have to be crossed in heterozygosis so that after having injected numerous pups, most of them were killed by the mothers and the expression of the virus was minimal. We have also generated lentivirus to overexpress p38a active in MKK6 KO cells. However, we found that the overexpression is toxic for the cell and in consequence the efficiency we obtained was too low to do any experiment.

In an attempt to answer the reviewer’s questions, we have decided to use the specifically KO mice in muscle for both MKK3 or p38a.

Our results suggest that hyperactivation of MKK3 in MKK6 KO cardiomyocytes is responsible for the mTOR activation and consequent cardiac hypertrophy. To try to elucidate the role of p38 in MKK3 hyperactivation as the referee asked, we generated the mice lacking p38a specifically in striated muscle. We found that lack of p38a induces the hyperactivation of MKK3 (Figure 5 —figure supplement 2). All these data suggest that the lack of p38a activation in MKK6 KO cardiomyocytes is responsible for the MKK3 hyperactivation and increased activation of p38g and p38d that induced cardiac hypertrophy.

**Author response image 2. sa2fig2:** MKK3 deletion attenuates p38g and mTOR activation and cardiac hypertrophy in MKK6-deficient hearts. (A) Heart weight to tibia length ratio in 9-week-old MCK-Cre (*n*=8), Mkk6^MCK-KO^ (n=10) and Mkk3/6^MCK-KO^ (*n*=10). 1-way ANOVA followed by Tukey’s multiple comparison test. **P*<0.05; ns: non-significant. Data are mean± SEM. (B) Immunoblot analysis of p38g phosphorylation and protein levels in heart lysates from 9-week-old Mkk6^MCK-KO^ and Mkk3/6^MCK-KO^ mice. (C) Immunoblot analysis of mTOR pathway activation by phosphorylation evaluation of its downstream target S6 in hearts from 9-week-old Mkk6^MCK-KO^ and Mkk3/6^MCK-KO^ mice.

9. The authors should check whether MKK3 interaction with p38γ/δ is enhanced in MKK6 KO mice. Proximity ligation assay would be useful for these experiments, also to get additional insights into the subcellular compartments in which these interactions may occur.

We agree with the reviewer that this is an interesting point as the interaction between the kinase and its substrate is an important mechanism for the regulation of phosphorylation. We have immunoprecipitated p38g/d and checked the interaction with MKK3 in heart lysates from MKK6 KO mice. We have found that lack of MKK6 increases the binding of these two kinases with MKK3 suggesting that this also may contribute to the p38g/d hyperactivation (Author response image 3). We agree that the proximity ligation assay would give us more information about the subcellular localization. However, due to the difficulty of this technical approach in tissue we cannot perform it and we decided to evaluate the question with an immunoprecipitation assay.

**Author response image 3. sa2fig3:** p38g/d immunoprecipitation in 9-week-old WT and *Mkk6^-/-^* mice indicating slight increased co-immunoprecipitation of MKK3 with p38g/d in hearts from MKK6-deficient mice.

10. p38 α/γ/δ activity should be measured in MKK6 KO mice.

To measure p38s activity in the heart of the MKK6 KO mice we studied the phosphorylation of their substrates. In agreement with the reduced p38a activation in *Mkk6^-/-^* hearts, we observed a reduction of its well-known substrate MAPAPK-2 (MK2) (Trempolec et al., 2017) (Author response image 4). Likewise, we observed and increased phosphorylation of the p38g/d substrate, SAP97/Dlg1 (Sabio et al., 2005) (Figure 9B), indicating an increased activity of these kinases in MKK6-deficient hearts.

**Author response image 4. sa2fig4:** p38a and p38g/d activity in MKK6-deficient hearts. (A) MAPKAPK-2 (MK2) phosphorylation and protein levels in hearts from 9-week-old WT and *Mkk6^-/-^* mice. (B) Evaluation of SAP97 phosphorylation at p38 specific motif Thr/Pro in hearts from 9-week-old WT and *Mkk6^-/-^* mice.

11. In the proposed mechanism (Figure 8), the authors suggest that p38α inhibits MKK3 in wild type mice. It is not clear how p38α regulates MKK3. Please address this issue.

We thank the reviewer for this excellent comment. We have generated the conditional p38a KO mouse in muscle (p38a^MCK-KO^) to evaluate this important point. We found that lack of p38a in the heart results in increased activation of MKK3 and MKK6 with higher levels of these proteins. These results are in agreement with previous studies that described that p38a can control their transcription (Ambrosino et al., 2003). In fact, associated with the reduced phosphorylation of p38a in hearts from MKK6 KO mice, we observed an increased expression of MKK3 at the protein and RNA level in these animals. Our results indicate that in the heart the activation of p38a promotes a negative feedback mechanism to regulate the activity of the upstream kinases at the expression and phosphorylation level. We have included these results in Figure 5 —figure supplement 2B/C.

12. The cause of death in aged systemic MKK6 KO mice should be understood. It is not clear whether these mice die because of cardiac abnormalities or through other non-cardiac issues. Do cardiomyocyte-specific MKK6 mice develop cardiac dysfunction and die prematurely?

We agree that this is an important question to address. To answer it, we evaluated whether, similar to the whole-body MKK6 KO, conditional KO mice lacking MKK6 specifically in cardiomyocytes (Mkk6^aMHC-KO^) also showed a reduced lifespan. We found that deletion of MKK6 restricted to heart is enough to induce premature death (Figure 4—figure supplement 2A). Additionally, this increased mortality was accompanied with cardiac dilatation and cardiac dysfunction, similar to the what we observed in *Mkk6^-/-^* mice, as revealed by echocardiographic analysis (Figure 4—figure supplement 2B/C). However, we agree with the reviewer that we cannot discard that, in the whole-body KO animals, dysfunction of other tissues may also contribute to the death of the mice. We have addressed this issue in the discussion.

13. The authors claim that activation of mTOR in MKK6 -/- mice is a direct consequence of p38γ/δ activation. However, this statement should be supported by additional evidence, also clarifying how p38γ/δ would regulate mTOR. DEPTOR phosphorylation and expression should be analysed, as performed in previous work from the same group (PMID: 26795633).

We thank the reviewer for the suggestion. We have now performed this analysis and found that, in agreement with increased phosphorylation of p38g/d and our previous work, DEPTOR is almost absent in MKK6 KO mice (Figure 7—figure supplement 1). Thereby, the evaluation of its phosphorylation is not possible to be assayed in this condition, since we would need to immunoprecipitate it for this aim. We were able to evaluate DEPTOR phosphorylation in our previous work because it was a different context. In p38g/d^-/-^ mice, there is a stabilization of the protein levels of DEPTOR due to the lack of p38g and p38d, and therefore we were able to address its phosphorylation.

References

Ambrosino C, Mace G, Galban S, Fritsch C, Vintersten K, Black E, Gorospe M, Nebreda AR. 2003. Negative Feedback Regulation of MKK6 mRNA Stability by p38α Mitogen-Activated Protein Kinase. *Mol Cell Biol* 23:370–381. doi:10.1128/mcb.23.1.370-381.2003

Brancho D, Tanaka N, Jaeschke A, Ventura JJ, Kelkar N, Tanaka Y, Kyuuma M, Takeshita T, Flavell RA, Davis RJ. 2003. Mechanism of p38 MAP kinase activation in vivo. *Genes Dev* 17:1969–1978. doi:10.1101/GAD.1107303

Sabio G, Arthur JSC, Kuma Y, Peggie M, Carr J, Murray-Tait V, Centeno F, Goedert M, Morrice NA, Cuenda A. 2005. p38γ regulates the localisation of SAP97 in the cytoskeleton by modulating its interaction with GKAP. *EMBO J* 24:1134–1145. doi:10.1038/SJ.EMBOJ.7600578

Trempolec N, Muñoz JP, Slobodnyuk K, Marin S, Cascante M, Zorzano A, Nebreda AR. 2017. Induction of oxidative metabolism by the p38α/MK2 pathway. *Sci Reports 2017 71*
**7**:1–15. doi:10.1038/s41598-017-11309-7

[Editors' note: further revisions were suggested prior to acceptance, as described below.]

The manuscript has been improved but there are some remaining issues that need to be addressed, as outlined below:Reviewer #3 (Recommendations for the authors):The authors made significant efforts to address my criticisms and the manuscript is significantly improved. However, some concerns remain and several data are still associative. In addition, important findings obtained during revision were not included in the actual version of the manuscript. In details:1) Figure 7 (rebuttal letter). Since MKK3 deletion does not rescue cardiac hypertrophy in MKK6 -/- mice, additional molecular players may be involved in addition to MKK3/mTOR signaling. Please discuss this aspect. Otherwise, the authors should evaluate cardiac hypertrophy as performed in Figure 2 and not only by the evaluation of heart weight.

We agree with the comment of the reviewer. As the reviewer points out, MKK3 deficiency in the context of MKK6 deletion in striated muscle produced a reduction in mTOR activation and attenuation of cardiac growth, but not was able to completely rescue the normal heart size. This result suggests that, apart from mTOR activation, MKK6 deficiency might affect other signaling pathways affecting cardiac hypertrophy. For example, p38γ/δ have been implicated in the regulation of the postnatal cardiac metabolism (Santamans *et al.*, 2021), which is another known direct cause of cardiac hypertrophy (Nakamura & Sadoshima, 2018). We have discussed this in the manuscript. Another possibility is that the incomplete recue was due to incomplete deletion of MKK3 in muscle as conditional mice did not always have full deletion of the target.

2) I understand the difficulties in generating mice with overexpression of p38α. However, in order to better dissect the proposed molecular mechanisms, the authors should test at least in vitro whether p38α overexpression decreases mTOR activation and cell hypertrophy in the presence of MKK6 deletion.

To address this question, we examined immortalized WT or MKK6-deficient mouse embryonic fibroblasts (MEFs). We found that MKK6-deficient MEFs showed mTOR pathway activation and increased cell size (Figure 7 —figure supplement 2 in the manuscript). Furthermore, lentiviral infection of MEFs with an active form of p38α (D176A/F327S) was able to reduce mTOR pathway activity as well as cell size (Figure 7 —figure supplement 2 in the manuscript).

We have included these results in the manuscript now.

3) The authors should discuss the new findings in the text.

We have addressed this question and discussed the new findings in the manuscript.

4) Western blots in the manuscript should be quantified.

We appreciate the suggestion of the reviewer and we have now included the quantification of the blots in the manuscript.